# Alternative TSS use is widespread in *Cryptococcus* fungi in response to environmental cues and regulated genome-wide by the transcription factor Tur1

Thi Tuong Vi Dang[1¤a], Corinne Maufrais[2], Jessie Colin[1,3], Frédérique Moyrand[1], Isabelle Mouyna[1], Jean-Yves Coppée[1], Chinaemerem U. Onyishi[4¤b], Joanna Lipecka[5], Ida Chiara Guerrera[5], Robin C. May[4], Guilhem Janbon[1] *

**1** Université Paris Cité, Institut Pasteur, Unité Biologie des ARN des Pathogènes Fongiques, Département de Mycologie, Paris, France, **2** Université Paris Cité, Institut Pasteur, HUB Bioinformatique et Biostatistique, C3BI, USR 3756 IP CNRS, Paris, France, **3** Ecole Pratique des Hautes Etudes, PSL Research University, Paris, France, **4** Institute of Microbiology and Infection and School of Biosciences, University of Birmingham, Birmingham, United Kingdom, **5** Université Paris Cité, SFR Necker INSERM US24/CNRS UAR3633, Proteomics Platform, Paris, France

¤a Current address: TUD Dresden University of Technology, Dresden, Germany
¤b Current address: Molecular Mycology and Immunity Section, Laboratory of Host Immunity and Microbiome, National Institute of Allergy and Infectious Diseases (NIAID), National Institutes of Health, Bethesda, Maryland, United States of America

* Guilhem.Janbon@pasteur.fr

**Data Availability Statement:** Raw and summarized sequencing data are available on GEO under accession number GSE237320 (RNA-seq,

## Abstract

Alternative transcription start site (TSS) usage regulation has been identified as a major means of gene expression regulation in metazoans. However, in fungi, its impact remains elusive as its study has thus far been restricted to model yeasts. Here, we first re-analyzed TSS-seq data to define genuine TSS clusters in 2 species of pathogenic *Cryptococcu*s. We identified 2 types of TSS clusters associated with specific DNA sequence motifs. Our analysis also revealed that alternative TSS usage regulation in response to environmental cues is widespread in *Cryptococcus*, altering gene expression and protein targeting. Importantly, we performed a forward genetic screen to identify a unique transcription factor (TF) named Tur1, which regulates alternative TSS (altTSS) usage genome-wide when cells switch from exponential phase to stationary phase. ChiP-Seq and DamID-Seq analyses suggest that at some loci, the role of Tur1 might be direct. Tur1 has been previously shown to be essential for virulence in *C. neoformans*. We demonstrated here that a *tur1Δ* mutant strain is more sensitive to superoxide stress and phagocytosed more efficiently by macrophages than the wild-type (WT) strain.

## Introduction

In eukaryotes, gene transcription begins at the core promoter by the assembly of a pre-initiation complex comprised of general transcription factors (TFs) that recruit the DNA-dependent

TSS-seq, DNA-seq, ChIP-Seq, DamID-Seq). Proteomic data has been submitted to ProteomeXchange via the PRIDE database under the accession number PXD045272.

**Funding:** TT VD was supported by the Pasteur-Paris University (PPU) International PhD Program. The funders had no role in study design, data collection and analysis, decision to publish, or preparation of the manuscript.

**Competing interests:** The authors have declared that no competing interests exist.

**Abbreviations:** CFW, calcofluor white; CuRE, Cu-responsive element; DMEM, Dulbecco's modified Eagle medium; FBS, fetal bovine serum; FDR, false discovery rate; HCD, higher-energy collisional dissociation; LUTI, long undecoded transcript isoform; MOI, multiplicity of infection; MTS, mitochondrial targeting signal; PI, phagocytic index; PIC, preinitiation complex; ROS, reactive oxygen species; SI, shape index; TAILS, terminal amine isotopic labeling of substrates; TF, transcription factor; TL, transcript leader; TSS, transcription start site; WT, wild type; uORF, upstream open reading frame.

RNA polymerase II and melt DNA to create the transcription bubble [1]. In recent years, different RNA sequencing strategies have been used in metazoans to define 2 types of core promoters characterized by specific chromatin structure, DNA sequence and distribution of transcription start site (TSS) [2]. These analyses have shown that TSS are by nature heterogeneous and organized in clusters. These TSS clusters can be sharp or broad, sharp ones being associated with TATA-box genes [1,3]. These studies also revealed that transcription can be initiated from multiple TSS clusters in most genes with a very dynamic pattern of usage [4–8] that can be specific to different cell types and development stages [6,9–11].

In fungi, although TSS sequencing data have been produced in several species [12–14], the analysis of TSS structure and usage is limited to the 2 model yeasts *Saccharomyces cerevisiae* and *Schizosaccharomyces pombe* [15–18]. Nevertheless, these studies also revealed a significant number of alternative TSS (altTSS) usage patterns, with some being specific to growth conditions or meiotic stages [12,19] and others revealed by mutation of genes encoding chromatin modifiers or remodelers [20–22].

*Cryptococcus neoformans* is a pathogenic basidiomycete yeast that is responsible for 180,000 deaths every year worldwide [23]. In addition to its ability to grow at 37°C, its main virulence factors are a polysaccharide capsule, the production of melanin and its ability to replicate in macrophages [24,25]. In recent years, we produced several sets of RNA-seq data that were used to produce detailed coding gene annotation of the genomes of several pathogenic *Cryptococcus* species [26–28]. Our analysis also revealed that nearly all the genes contain several short introns, which are essential for gene expression [29,30]. As expected, alternative splicing is prominent in *Cryptococcus*, although its impact on proteome structure is limited primarily to regulating gene expression [27]. More recently, we produced TSS-seq and 3UTR-seq data from *C. neoformans* and its sibling species *Cryptococcus deneoformans* grown under 4 conditions (i.e., exponential phase and stationary phase at 30 and 37°C) [14]. We used this dataset to re-annotate the transcript leader (TL) and 3′ UTR sequences in these species. Our analysis revealed that, in contrast to *S. cerevisiae*, *Cryptococcus* TL sequences frequently contain upstream open reading frames (uORFs). These yeasts use the strength of a Kozac-like consensus to determine translation start codon usage, thus regulating both gene expression and protein localization [14]. This analysis also revealed thousands of additional TSS clusters associated with coding genes, suggesting that alternative TSS (altTSS) usage might be widespread in *Cryptococcus* [14]. This hypothesis was supported by several recent studies reporting gene-specific examples of altTSS usage regulation in these yeasts. For instance, Pum1, an RNA-binding protein, is known to positively regulate the expression of *ZNF2*, a master regulator of filamentation and virulence in *C. neoformans* [31]. Under filamentation-inducing conditions, the TF Znf2 favors *PUM1* transcription from a downstream TSS, which excludes from the transcript the Pum1-binding site that is normally found in the TL. This shorter form being immune to the negative autoregulation, this leads to the accumulation of the Pum1 protein which in turn activates *ZNF2* expression [31]. Moreover, when exposed to UV light, *C. neoformans* switches off TSS at the *UVE1* gene and uses an upstream altTSS. This promotes the transcription of a longer mRNA coding for the mitochondrial isoform of the DNA damage repair endonuclease Uve1, thus protecting the mitochondrial genome from potentially lethal UV-induced DNA damage [32]. Finally, a recent study reports that, under copper-limited conditions, *C. neoformans* cells promote the usage of a downstream altTSS at both *SOD1* and *SOD2* superoxide dismutase genes, thus regulating transcript stability and protein subcellular localization of these proteins, respectively [33].

In the present study, we re-analyzed these *Cryptococcus* TSS-seq data to first characterize the structure of a genuine TSS cluster in these yeasts. We then described alternative TSS usage and demonstrated that it represents a major means to regulate transcriptome and proteome

structure in these fungi. More importantly, we screened a TF mutant collection to identify genes regulating altTSS usage in *C. neoformans*. We showed that the transcription factor Tur1 is necessary for genome-wide altTSS usage regulation during the exponential to stationary phase transition. We also performed ChiP-Seq and DamID-seq analyses to study the binding of Tur1 at the regulated loci. Finally, we showed that Tur1 regulates superoxide stress resistance and interaction with macrophage linking altTSS usage and virulence in this major fungal pathogen.

## Results

### TSS-cluster characterization in *Cryptococcus*

We previously used TSS-seq data obtained from cells grown under 4 conditions in triplicate to construct 12 GFF files defining the coordinates of several thousand TSS clusters associated with coding genes in 2 *Cryptococcus* species [14]. To evaluate alternative TSS usage, we aimed to construct a TSS cluster reference GFF file. We first thought to merge the 12 condition-specific GFF files originally constructed. However, this simplistic strategy resulted in a very poor result. Firstly, most of the resulting merged TSS were doubtful since they were only 1-bp wide. Secondly, at several loci the TSS cluster size and number varied between conditions. This merging operation also resulted in the definition of very large TSS (more than 100-bp wide) with probably poor biological relevance (see S1 Fig). This result suggested that the strategy previously used to define the TSS clusters [14] was not optimal and resulted in the identification of a number of doubtful TSS clusters. To better define the characteristics of genuine TSS clusters in *C. neoformans*, we reasoned that when only 1 TSS cluster is found upstream of the annotated ATG (aATG) of a well-expressed gene, this cluster should be genuine. We thus first considered only the 50% most expressed genes in exponential phase 30°C and for which a single TSS cluster has been originally identified upstream of the aATG ($n = 761$). We then performed subclusterization analysis (see Material and methods) and characterized the structure of the resulting clusters using 2 parameters: size and thermodynamic entropy-related shape index (SI), which is a measure of the heterogeneity of the TSS cluster [34]. Finally, we kept only the TSS clusters with both sizes and SI values comprised between the 2.5% and 97.5% quantiles for further analysis ($n = 719$). Plotting the size and SI of this set of newly defined TSS clusters with the MixtureInf [35]-based statistical test revealed a bimodal distribution defining 2 categories: the Sharp clusters characterized by a lower size and high SI values and the Broad TSS clusters having the converse characteristics (Fig 1A and 1B).

MEME Suite software 5.4.1 [36] de novo pattern discovery mode was used to examine the region going from −100 to +30 relative to the major position within each TSS cluster to identify 6 enriched motifs (Fig 1C). Notably, our analysis identified a TATA motif positioned −40 to −35 bp upstream of the TSS. We noticed that genes with a sharp TSS cluster are enriched within the TATA-box containing genes, as previously observed in some other model organisms [1] (Fig 1C). Overall, 23.5% of the Sharp TSS clusters are associated with a TATA-box containing genes compared to only 11.7% of the Broad TSS clusters and 14.5% of both groups of TSS clusters ($\chi^2$ test, *p*-value <0.05). The distance between the TATA-box and the TSS in *Cryptococcus* is comparable with that reported in metazoans (28 to 33 bp) and in *S. pombe* (25 to 32 bp) [2,3]. This short distance suggests that *C. neoformans* uses a classic model of transcription initiation in which the preinitiation complex (PIC) assembles on TATA-box and loads the RNAPII on a TSS located in close proximity. In contrast, *S. cerevisiae* as well as most hemiascomycetes use a scanning model of transcription initiation in which the PIC scans the sequence downstream to find TSSs which results in longer and more variable distances between the TATA-box and the TSS (40 to 120 bp) [3].

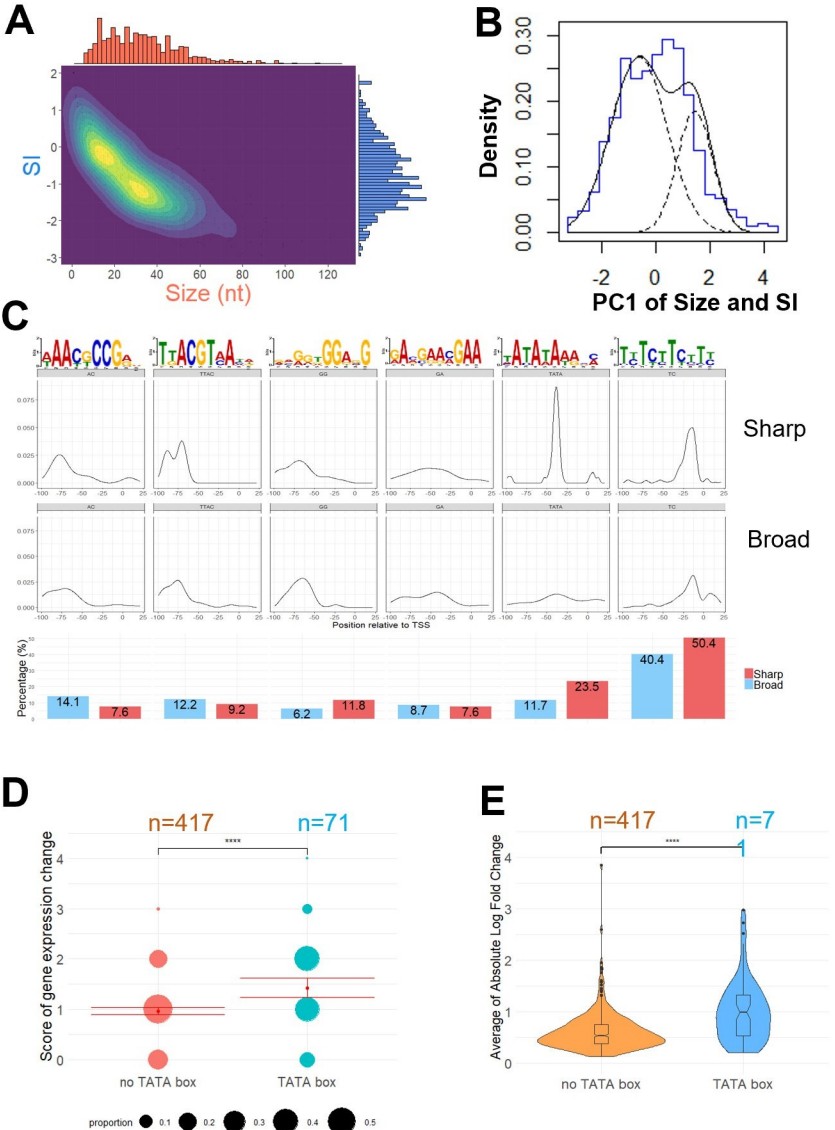

**Fig 1. TSS cluster classification, motif enrichment, and the relationship with gene expression.** (A) The distribution of TSS clusters in terms of cluster width (size) and SI is represented as a 2D density plot in which size and SI distribution of TSS clusters display a bimodal pattern. Corresponding histogram of size and SI are projected on the x-axis and y-axis, respectively. (B) Histogram (bold blue line) and density plot (bold black line) of the first principal component (PC1) of size and SI. Two subpopulations of PC1 are detected by statistical test using "MixtureInf" R package and represented as the theoretical density plot (doted black line). (C) Upper panel: Enrichment and position relative to the major position within the TSS cluster of 6 detected motifs within the promoter sequence. Lower panel: Percentage of presence of each motif in Sharp and Broad clusters associated genes. (D) Score of gene expression change for TATA-box-harboring genes (pink) or devoid of TATA-box (blue) are represented as dots. Dot size is proportional to the percentage of each score value in y-axis. Sample sizes *n*, mean (red dots), and confidence interval (red lines) of the mean are indicated for both groups. Significance was computed using Wilcoxon rank sum test, *p*-value <0.0001. (E) Magnitude of gene expression change for TATA-box-harboring genes and genes devoid of TATA-box is depicted as violin plots and box plots. Average of Absolute Log Fold Change in 4 growth conditions compared to WT (main text) is on y-axis. Sample sizes n are indicated for both groups. Significance was computed using Wilcoxon rank sum test, *p*-value <0.0001. The data underlying this figure can be found in S1 Data. SI, shape index; TSS, transcription start site; WT, wild type.

We performed the same analysis using the TSS-seq data obtained from cells grown in the 3 other conditions (i.e., exponential phase 37°C; stationary phase 30°C and 37°C). Even though the gene set considered was different each time, similar bimodal distribution of sizes and SI as well as motif enrichments and positions were observed, suggesting general features in *C. neoformans* (S2 and S3 Figs). Previous studies in other organisms indicated that regulated genes are associated with the presence of a TATA-box, whereas constitutive ones tend to be devoid of TATA-box [2,5,8,34,37]. To determine if this phenomenon is conserved in *C. neoformans*, we used or produced spiked in RNA-seq data from *C. neoformans* cells grown in diverse conditions (i.e., exponential phase at 30°C in the presence of either fluconazole (15 μg/ml) or SDS (0.01% w/v), exponential phase at 37°C or stationary phase at 30°C) [14] to perform differential gene expression analysis using RNA-seq data produced from cells growing in exponential phase at 30°C as the reference. We then used a variability score as a proxy to evaluate the degree of gene expression variability. A gene not regulated in any of these 4 alternative conditions will have a score of 0, whereas a gene regulated by all these modifications of culture conditions will have a score of 4. Our results revealed that genes harboring a TATA-box in their promoter are more prone to be regulated than the genes of the control group (devoid of TATA-box) (Wilcoxon rank sum test, *p*-value <0.0001) (Fig 1D). TATA-box containing genes also display a higher absolute log fold-change between conditions than genes devoid of TATA-box (Wilcoxon rank sum test, *p*-value <0.0001) (Fig 1E). As said above, we identified 5 additional enriched motifs proximal to the *Cryptococcus* TSS. Some of these motifs like "AAYKCCG" were more commonly associated with broad TSS some other as "GGRNG" were more commonly associated the with sharp TSS. Yet, their functionally remains to be studied.

Given that Sharp TSS cluster associated genes are enriched in TATA-boxes, it is no surprise that these genes are also more prone to be regulated and with a higher magnitude than Broad cluster associated genes (Wilcoxon rank sum test, *p*-value <0.05) (Fig 2A). To figure out which feature, the presence of TATA-box or the TSS cluster shape, is the true determinant for gene expression variability, we conducted stratification analyses. We compared absolute log fold-change between Sharp and Broad cluster-associated genes within the TATA-box-harboring genes and within the no-TATA-box genes. In both groups, the difference between Sharp and Broad cluster is no longer detected, suggesting that this higher magnitude of gene expression change in Sharp cluster is most likely due to higher percentage of TATA-box containing genes in this group (Fig 2B). Meanwhile, TATA-box containing genes display higher absolute log fold-change between conditions compared to the genes without TATA-box regardless of the TSS cluster shape (Wilcoxon rank sum test, *p*-value <0.0001) (Fig 2C) confirming the presence of TATA-box as the decisive factor to explain gene expression variability in *C. neoformans*. Overall, this re-analysis of TSS-seq data revealed the existence of 2 types of TSS clusters in *Cryptococcus*. Sharp TSS cluster-containing genes are more often associated with a TATA-box and more prone to be regulated in response to a modification of the environment than the Broad TSS cluster containing genes. These results are in good agreement with data published on several other organisms [1,12,16,38,39] suggesting that our TSS cluster definition approach is valid.

## Alternative TSS cluster identification in *Cryptococcus*

To evaluate alternative TSS usage in *Cryptococcus*, we applied the same subclusterization procedure to all previously identified TSS clusters [14]. We then merged the overlapping TSS clusters obtained from the 4 growth conditions in triplicate. The resulting TSS cluster reference GFF file defines the genomic coordinates of 7,213 TSS clusters associated with 4,931 coding genes in *C. neoformans* (72.6% of the coding genes) (S4 Table). We defined a TSS cluster as

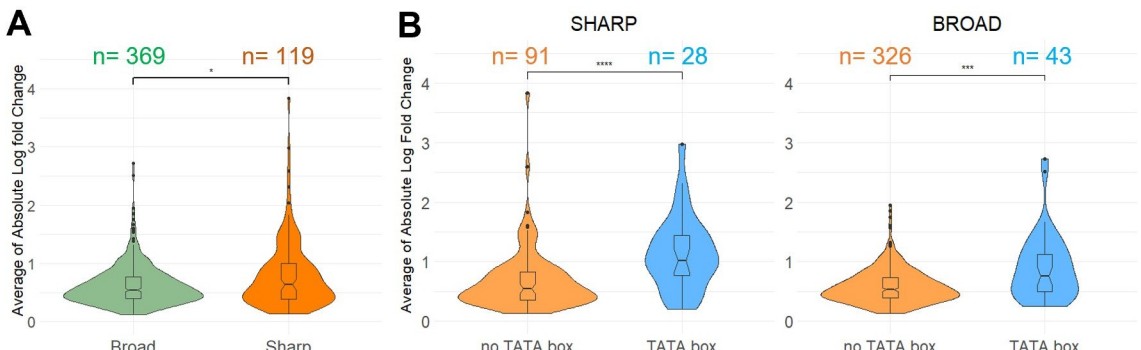

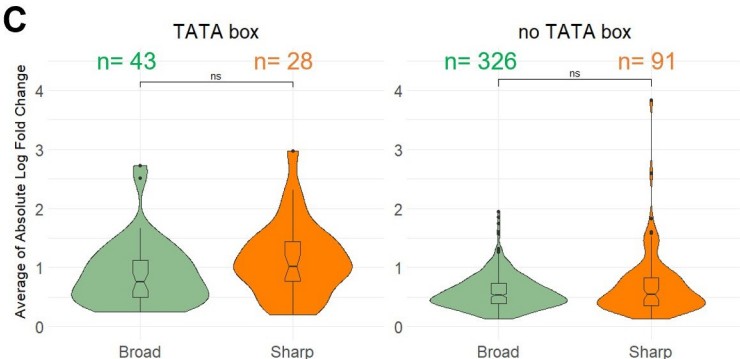

**Fig 2. Relationship between TSS cluster shape, TATA-box and gene expression dynamic.** (A) Magnitude of gene expression change of Sharp cluster genes and Broad cluster genes is depicted as violin plots and box plots. Average of Absolute Log Fold Change in 4 growth conditions compared to WT (main text) is on y-axis. Sample sizes *n* are indicated for each group. Significance was computed using Wilcoxon rank sum test, *p*-value <0.05. (B) Stratification analysis comparing TATA-box-harboring genes and TATA-box-devoid genes within those having Sharp clusters (left panel) and those with Broad clusters (right panel). Sample sizes n are indicated for each group. Significance was computed using Wilcoxon rank sum test, *p*-value <0.0001 for both comparisons. (C) Stratification analysis comparing genes with Sharp clusters and genes with Broad clusters within TATA-box-harboring genes (left panel) and TATA-box-devoid genes (right panel). Sample sizes *n* are indicated for each group. Significance was computed using Wilcoxon rank sum test, *p*-value >0.05 for both comparisons. The data underlying this figure can be found in S1 Data. TSS, transcription start site; WT, wild type.

annotated TSS clusters if it satisfies all these 3 criteria: it is the most upstream cluster within a gene, the distance between the 5′ boundary of the cluster and the annotated 5′ extremity of the gene does not exceed 50 bp, and the 3′ border of the cluster is at least 20 bp upstream of the annotated ATG. Every other TSS clusters were considered as alternative TSS clusters. Here, we identified 2,431 genes associated with 2 or more TSS clusters. In total, 3,817 TSS clusters were considered as alternative. We performed the same analysis using the *C. deneoformans* data and defined 4,064 alternative TSS clusters associated with 2,581 genes (S4 Table).

## altTSS within the TL sequence

A large proportion of these alternative TSS clusters (37% (*n* = 1,405) in *C. neoformans* and 31% (*n* = 1,256) in *C. deneoformans*) are positioned within the TL sequence. Their usage regulates the length of the TL sequence (Fig 3) and can result in the inclusion or exclusion of protein binding sites, secondary structures or uORFs, potentially impacting mRNA subcellular localization, stability, or translation efficiency [14,15,40]. We previously reported that *Cryptococcus* TL sequences are very rich in uORFs and that uORF containing mRNAs are more prone to be degraded by the NMD pathway than uORF-free mRNAs, thus reducing translation

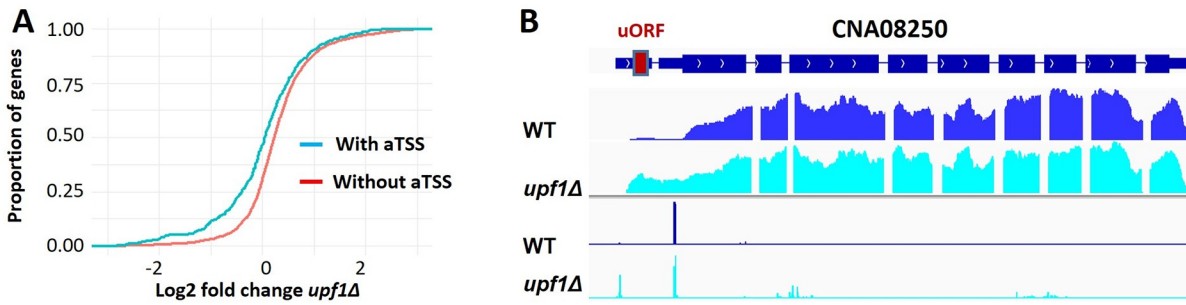

**Fig 3. altTSS usage regulates the presence or absence of uORFs within the mRNA impacting its sensitivity to NMD.** Among 3,055 uAUG containing genes in *C. deneoformans*, genes having alternative TSS within their TL sequence are more resistant to NMD. (A) Cumulative distribution plot of Log fold change between *upf1Δ* mutant and WT at 30°C are generated for genes which use an altTSS at 30°C (blue line) and genes that do not use an altTSS at 30°C (red line). A one-sided Kolmogorov–Smirnov test shows that these differences are significant (*p*-value = $1.95 \times 10^{-10}$). (B) IGV visualization of RNA-seq and TSS-seq at the CNA08250 locus. The gene CNA08250 possesses an alternative TSS that skips the uORFs of the transcript leader sequence. At 30°C, the alternative TSS is prominently used, resulting in the short mRNA isoform devoid of uORF. The data underlying this figure can be found in S1 Data. TL, transcript leader; TSS, transcription start site; WT, wild type; uORF, upstream open reading frame.

efficiency [14]. Here, we identified 542 and 633 genes in *C. neoformans* and *C. deneoformans*, respectively, for which the sequence between the annotated TSS cluster and the alternative TSS cluster located in the TL contains 1 or several uORFs. In these genes, alternative TSS usage can regulate the presence or absence of uORFs within the mRNA. In these cases, we reasoned that the production of a short transcript devoid of uORFs would be a way to stabilize it by preventing its degradation by the NMD pathway. Accordingly, among the genes containing 1 uORF within their TL sequence, the ones containing an alternative TSS potentially skipping it are less prone to be up-regulated by the deletion of the major NMD factor *UPF1* than those devoid of alternative TSS within their TL sequence (Fig 3A) (Wilcoxon rank sum test, *p*-value = $1.246 \times 10^{-10}$). For instance, usage of an alternative TSS within the TL sequence of the CNA08250 locus skips 11 uORFs and results in the production of a short isoform immune to the NMD, whereas the long isoform can only be revealed by the deletion of *UPF1* (Fig 3B).

## altTSS close to the annotated start codon

We identified 220 alternative TSS clusters associated with 213 genes in *C. neoformans* (208 TSS clusters and 199 genes in *C. deneoformans*) positioned between −20 and +90 nt from the aATG, when counting bases on the spliced mRNA. The usage of each of these altTSSs results in the transcription of an mRNA which can code a protein truncated from its N-termini, thereby potentially lacking an N-terminal targeting motif. Accordingly, DeepLoc 2.0 [41] analysis predicts that whereas for 169 of these genes the long isoform has a defined subcellular localization, for 87 of them the usage of an altTSS is predicted to result in the production of a shorter isoform with a different subcellular localization (S5 Table and Fig 4). For instance, we identified an altTSS associated with the gene *MAE102* (CNAG_06374) that encodes a putative mitochondrial malate dehydrogenase (Fig 5A). Depending on the TSS used, the predicted translated protein contains a mitochondrial targeting signal (MTS) or not. Interestingly, the usage of this altTSS is regulated by the growth condition, the long isoform encoding the putative mitochondrial protein being transcribed in stationary phase at 30°C, whereas the exponential phase condition triggers mostly the transcription of the short isoform predicted to code a protein devoid of MTS. Accordingly, a C-terminal mNeonGreen-tagged version of Mae102 protein revealed a regulated localisation of this protein by the phase of growth. It seems to accumulate into mitochondrial-like particles in stationary phase (Fig 5B), whereas in

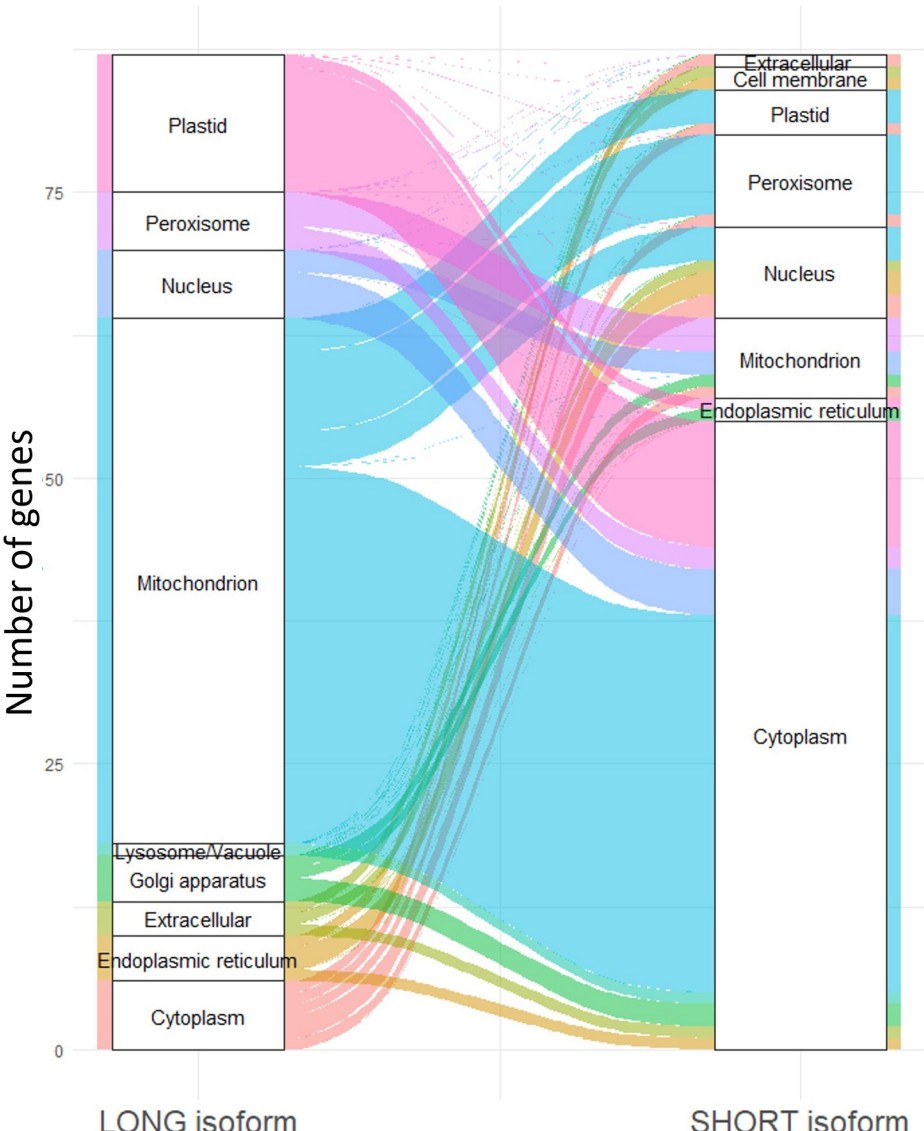

**Fig 4. altTSS usage can result in the production of proteins shorter than the annotated ones, lacking N-terminal targeting motif and targeted to a different organelle.** Alluvial diagram depicting the predicted localization of long and short protein isoforms as predicted by DeepLoc 2.0 [41]. TSS, transcription start site.

exponential phase it localizes in the cytosol and the nucleus (Figs 5B and S4). Moreover, mutation of the first ATG into CGT (M1R) or deletion of the sequence between the 2 ATGs (MTSΔ) increase the percentage of cells having a nucleus localization of the protein in exponential phase. In stationary phase, the Mae102 protein mitochondrial-like localization seems to be lost confirming the functionality of the MTS sequence. Overall, these results are consistent with the alternative localization of the Mae102 isoforms depending on the alternative TSS usage regulated by the growth phase.

## altTSS downstream of the start codon

The 2,181 remaining TSS clusters (2,576 in *C. deneoformans*) are positioned at least 90 bp after the annotated start codon and promote the transcription of shorter RNAs than the annotated

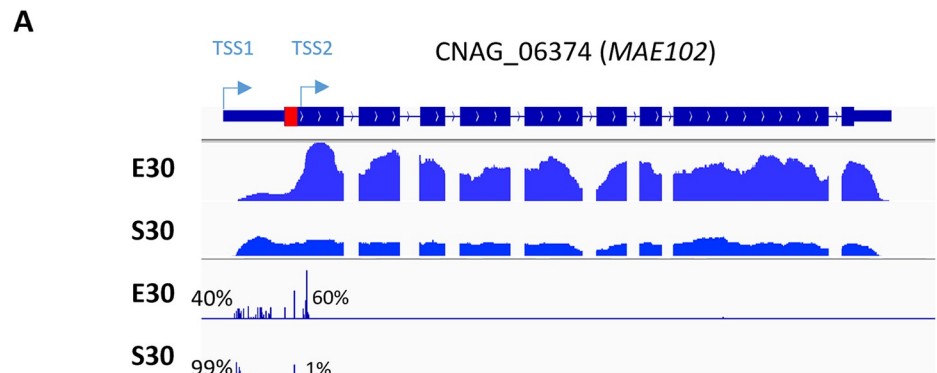

**A**

**B**

**Fig 5. Mae102 subcellular localization is regulated through alternative TSS usage.** (A) IGV visualization of RNA-seq and TSS-seq at the *MAE102* locus. In exponential phase at 30°C cells mainly use an altTSS (TSS2) located 19 bp downstream of the annotated ATG. The shorter mRNA produces a protein isoform lacking the MTS (red box) as predicted by DeepLoc 2.0 and MitoFates [41,98]. In stationary phase at 30°C, the annotated TSS (TSS1) is mainly used to produce a full-length transcript that is translated in a protein containing an MTS at its N-terminal. In exponential phase at 30°C, usage of the downstream TSS2 produces a protein expected to go mainly to cytosol. The percentages indicate the

proportion of each TSS cluster used in each condition. Detailed numbers are given in S8 Table. Alternative usage of the 2 TSS clusters (B) DIC and fluorescent microscopy images of *C. neoformans* cells expressing a Mae102-mNeonGreen fusion protein (green) in different mutants: WT, *mae102-MTSΔ*, *mae102-M1R*, and *tur1Δ*. Mitochondria are stained using MitoTracker (red). The percentages refer to the proportion of cells with a Mae102 nuclear localization when are grown under exponential phase. MTS, mitochondrial targeting signal; TSS, transcription start site; WT, wild type.

ones (S4 Table). We named this last category of transcripts TRASS for Transcript Resulting from Alternative Start Sites. One example of TRASS is given in Fig 6. At the CNC03460 locus in *C. deneoformans*, cells growing in exponential phase at 30˚C use mainly an altTSS located within the sixth intron of the gene and positioned within a CDS-intron 1,395 bp downstream of the annotated TSS. In contrast, cells in stationary phase mainly use the annotated altTSS as confirmed by northern blot analysis (Fig 6B).

Although the function of these RNAs was unknown, we noticed that nearly all the TRASS have coding capacity. Overall, 1,523 potential new proteins could be encoded by these RNAs in *C. neoformans*. Most of these new proteins would be in frame with the annotated ones and would thus be completely ignored though classical proteomic analysis. To gain insights into the coding capacities of the TRASS, we performed N-terminomic analysis [42] using proteins extracted from cells growing in stationary phase at 30˚C. We identified 844 peptide sequences corresponding to N-terminal sequences associated with 12% of the coding genes ($n$ = 810) in *C. neoformans*. As expected, most of the N-terminal peptides (97%; $n$ = 818) correspond to the annotated N-terminal sequence of 784 proteins (Table A in S6 Table). Also as expected, we identified 19 peptides corresponding to 19 proteins presenting an alternative N-terminal sequence produced from an altTSS close to the annotated ATG (Table B in S6 Table). Finally, 7 N-terminal sequences are likely the products of translation of 7 TRASS (Table B in S6 Table). A striking case is the gene CNAG_04307 (*URO1*) coding Urate oxidase for which we identified a peptide that is out-of-frame with the main protein. Of course, these represent only a few examples of peptides translated from these RNAs and it is probable that a large part of the TRASS are lncRNAs. Nevertheless, N-terminomic analysis, although powerful and efficient in selecting protein N-termini, is not powerful enough for low abundance alternative N-termini suggesting that much more TRASS encoded peptides are produced in these cells.

## Alternative TSS usage in *Cryptococcus*

To further explore the dynamics of alternative TSS usage in *Cryptococcus*, we evaluated how each TSS is employed depending on the growth phase (Exponential or Stationary) and temperature (30˚C or 37˚C). TSS cluster usage was evaluated considering an"expression level" in each

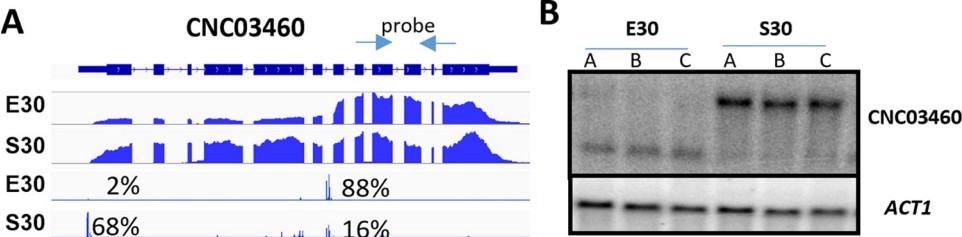

**Fig 6. altTSS usage regulates the production of TRASS in *Cryptococcus*.** (A) IGV visualization of RNA-seq and TSS-seq at the CNC03460 locus when *C. deneoformans* cells were cultivated at 30˚C under exponential (E30) or stationary (S30) phase, respectively. The percentages indicate the proportion of each TSS cluster used in each condition. Detailed numbers are given in S8 Table. (B) Northern blot validation of the alternative RNA molecule production. For CNC03460, the probe was amplified using primers specific of the second half of the gene (arrows). The actin gene (*ACT1*) was used as reference. TSS, transcription start site.

replicate of each condition measured by the number of TSS-seq reads associated within a given TSS cluster. These numbers were then normalized by the total amount of TSS-seq reads aligned to each coding gene. To limit the potential bias associated with this type of normalization, we limited our analysis to genes for which we could count at least 20 TSS reads in each considered condition. Overall, we considered 3,648 TSS clusters belonging to 1,366 genes in *C. neoformans* under 4 growth conditions (4,311 TSS clusters belonging to 1,621 genes in *C. deneoformans*) for this analysis. We confidently identified 1,478 altTSS usage regulations by growth condition in 627 *C. neoformans* genes (Fig 7A and S7 Table). Alternative TSS usage seems to be more dynamic in *C. deneoformans* with 2,369 significant regulations in 1,070 genes. Interestingly, 25% of the altTSS usage regulations are specific to the considered comparison (Fig 7A). Explicit examples of these regulations are given on the Fig 7B. At the gene *PKP1* (CNAG_00047), 2 TSS are used alternatively depending on the phase of growth. In stationary phase at 30˚C, an altTSS within the second intron is mainly used and promotes the transcription of a TRASS. Strikingly, we also identified *SOD1* and *SOD2* genes as regulated by altTSS usage during this phase transition (S5 Fig). In that case, the pattern of altTSS usage in stationary phase is similar to the one previously reported in copper-limiting condition using gene-specific experiments [33]. Alternative TSS usage at these 2 genes has been reported to be regulated by copper and is dependent on the transcription factor Cuf1 which can recognize a Cu-responsive element (CuRE) found in their promoters [33]. It is possible that the stationary phase condition somehow mimics the copper shortage as used by the Thiele laboratory thus altering altTSS usage at these loci in the same way. We also observed regulation of altTSS by the temperature. For instance, at the locus CNAG_00812, the usage of a downstream TSS is favored in stationary phase at 37˚C thus promoting the production of a TRASS. Moreover, at the loci CNAG_01272 and CNAG_03239, a change in temperature in stationary and exponential phase, respectively, alters the usage of an altTSS located within the TL sequence (Fig 7B). Our original idea was to compare the results obtained with the 2 studied sibling species of *Cryptococcus*. We observed an apparent low conservation of these regulations with only 16.21% and 10.12% of the altTSS usage regulations observed in *C. neoformans* upon the transition of the growth phase and temperature change, respectively, conserved in *C. deneoformans* (S4 Table). However, visual examination of some of the "non-conserved" events revealed that the apparent non-conservation was mainly due to the poor level of one or the other TSS in one species which did not pass the different thresholds of our bioinformatic pipelines (S6 Fig). Indeed, in some cases, the position of the altTSS is conserved but regulation is reversed. In others, the altTSS exists but its position is not conserved (S6 Fig). These observations suggest that the degree of conservation of these regulations might be more significant than they appear at first glance.

## Tur1 regulates altTSS usage at the *PKP1* gene

Our analysis revealed a widespread regulation of altTSS usage in *Cryptococcus* regulating both transcriptome and proteome structures in response to modifications of the growth condition. Although the mechanisms associated with these regulations remain unknown, the specificity of some of these regulations suggests that alternative TSS usage is mediated by precise regulators. We reasoned that at least some of these regulators might be TFs as previously observed in mammals and in yeast [9,10,33,43]. To identify altTSS usage regulators in *Cryptococcus*, we screened a library of 155 TF mutant strains [44] using an RT-qPCR assay with primers specific to the long or both the long and the short isoforms identified at the *PKP1* gene (CNAG_00047). This gene encodes a protein that shares homology with the *S. cerevisiae* mitochondrial protein kinase Pkp1 [45]. In *C. neoformans*, *PKP1* transcription can start at 2 altTSS

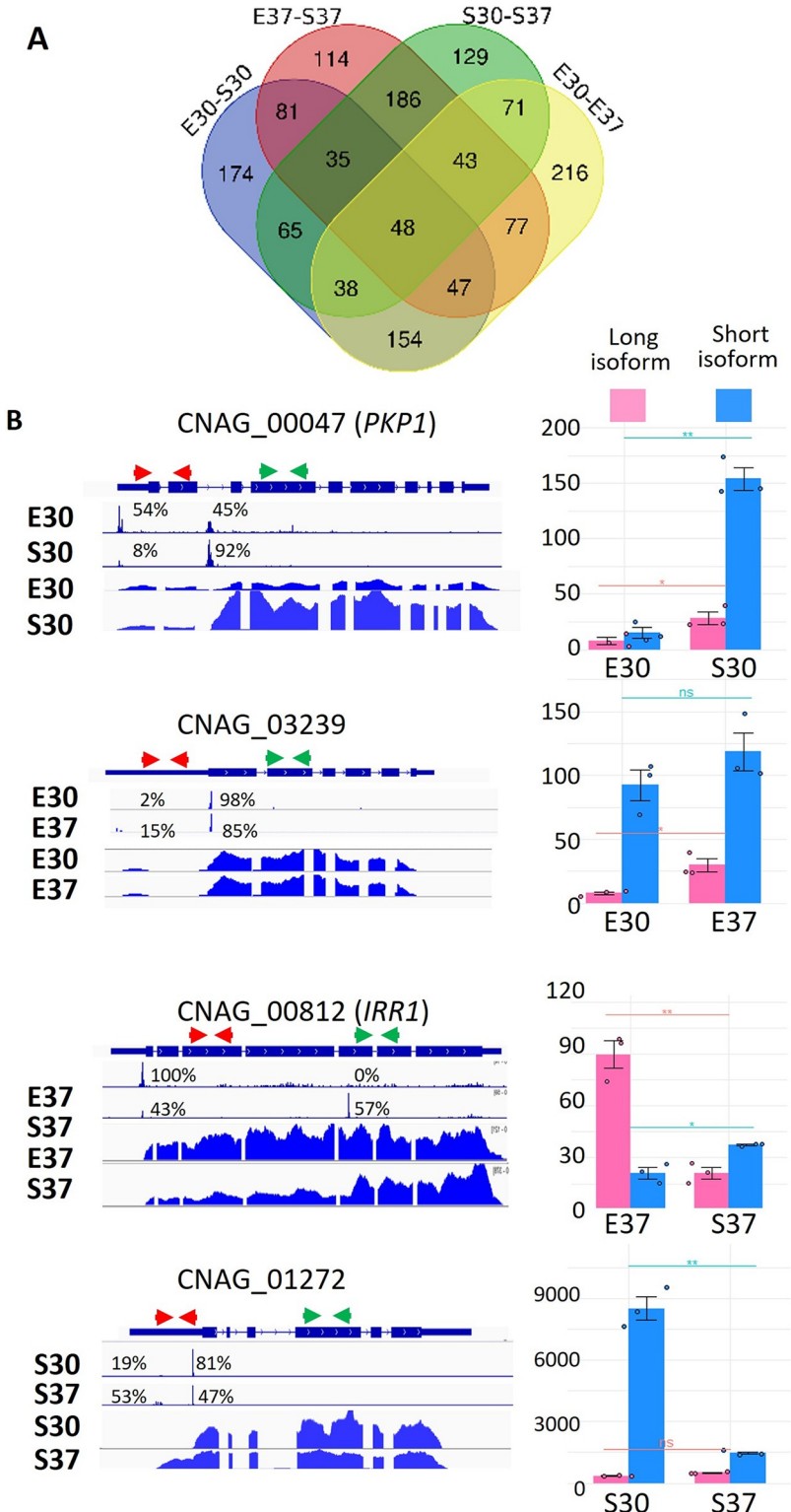

**Fig 7. Differential analysis of altTSS usage in response to the growth phase and temperature changes.** (A) Venn diagram illustration the overlap between the altTSS usage regulation (FDR < 0.05) in 4 conditions: exponential phase (E30), stationary phase (S30) at 30°C, exponential phase (E37), and stationary phase (S37) at 37°C in *C. neoformans*. (B) IGV visualization of RNA-seq and TSS-seq of examples altTSS usage regulated by the temperature and the growth phase. The percentages indicate the proportion of each TSS cluster used in each condition. Detailed numbers are given

in S8 Table. qRT-PCR confirmation of these regulations was performed using primers specific of one or both isoforms. Y-axis shows 1,000*fold-change compared to the number of *ACT1* mRNA molecules. Experiments were performed in biological triplicates and technical duplicates. Error bars are shown median +/− standard deviation. Green and red arrows indicate the position of the primers used for these experiments. FDR, false discovery rate; TSS, transcription start site.

clusters with different expression patterns in exponential and stationary phase at 30˚C (Fig 7B). RNA was extracted from the wild-type (WT) strain and the 155 TF mutant strains grown in stationary phase at 30˚C. The level of the long and the short mRNA isoforms was then evaluated by RT-qPCR. We identified a single mutant strain altered for altTSS usage at this locus. This mutant strain displayed no growth defect and is mutated at the locus CNAG_05642 encoding a zinc cluster TF previously designated Fzc37 [44]. Sequence conservation analysis revealed that this TF is basidiomycete specific with no predicted function. In this mutant strain, the short isoform is strongly down-regulated in stationary phase compared to the WT strain (*t* test *p*-value <0.01), whereas the expression of the long isoform is not altered (*t* test *p*-value >0.05) (Fig 8A). We therefore renamed this gene *TUR1* (TSS Usage Regulator 1). As expected, complementation of the *tur1Δ* mutation restored the WT expression of the short isoform (Fig 8A). We confirmed the role of Tur1 in the regulation of altTSS usage at the *PKP1* gene by constructing a conditional mutant in which a 2xFlag-CBP tagged version of *TUR1* is expressed under the control of the *C. neoformans GAL7* promoter [46] (Fig 8C). As expected, the expression of the *PKP1* short isoform was only observed in stationary phase when the cells were grown with galactose as the sole source of carbon. No regulation was observed when cells were grown in glucose (Fig 8B), a condition in which the *GAL7* promoter is repressed and no Tur1 protein is produced (Fig 8B).

## Tur1 regulated transcriptome structure in *C. neoformans*

To get more insights into the role of *TUR1* in *C. neoformans* biology, we performed spiked-in RNA-seq experiments using RNA extracted from *tur1Δ* cells grown in either stationary or exponential phase. In the WT background, the transition between exponential phase to stationary phase is associated with a strong decrease of the transcriptome size with 82% (*n* = 5,582) of the genes down-regulated at least 2-fold (Fig 9A and S8 Table). Our estimate is that the number of RNA molecules is reduced by 3.8-fold per genome during this phase transition. This is similar to that observed previously in *S. cerevisiae*, in which the overall transcription rate is reduced about 3 to 5 times during the exponential to stationary phase transition [47]. In the *tur1Δ* mutant strain, this slowdown of the metabolism is less marked than in the WT strain, and we measured only a 1.5-fold reduction of the total number of RNA molecules per genome and identified only 27% (*n* = 1,814) of the genes down-regulated by at least 2-fold (Fig 9A and S8 Table). Interestingly, nearly 97% of these down-regulated genes in the mutant strain were also down-regulated in the WT background (Fig 9B). GO-term analysis showed that this set of conserved down-regulated genes is enriched in genes coding translation and ribosome biosynthesis-linked functions. In contrast, the set of genes down-regulated in the WT but not the *tur1Δ* strain is enriched in genes coding proteins linked to transcription regulation suggesting that this TF is necessary for such regulation.

This striking effect of *TUR1* deletion on transcriptome structure regulation during exponential to stationary phase transition suggested that this TF could have a genome-wide effect on TSS regulation. To discover whether Tur1 would regulate altTSS at any other locus, we performed TSS-seq in the *tur1Δ* mutant strain using cells grown in exponential and stationary phase at 30˚C. Here again, in a very conservative way, we restricted our analysis to only 806

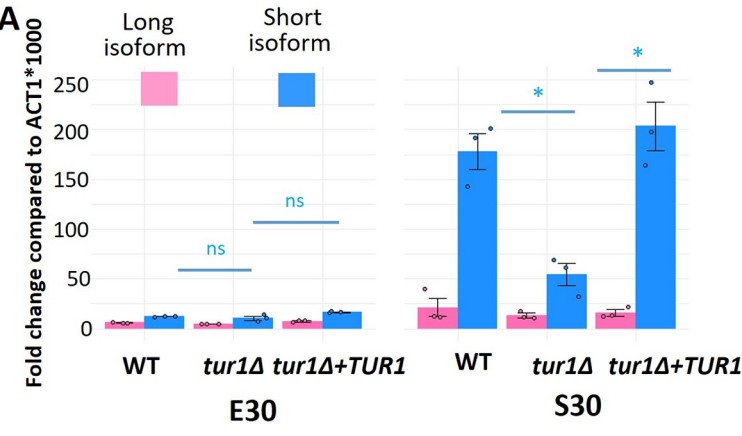

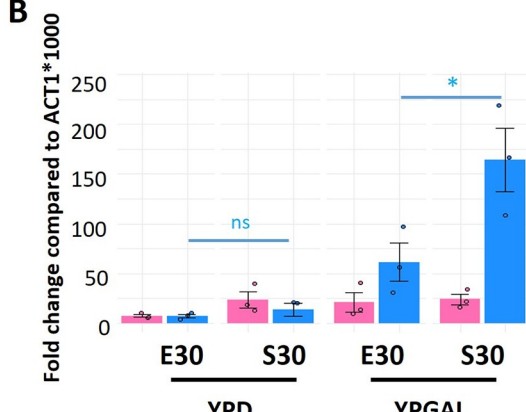

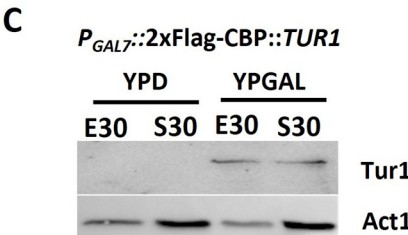

**Fig 8. Tur1 regulates the altTSS usage at *PKP1* during the transition from exponential phase to stationary phase at 30˚C.** (A) RT-qPCR measuring the level of the long and short mRNA isoforms of *PKP1* in WT, *tur1Δ*, and the complemented *tur1Δ+TUR1* strains grown in stationary or exponential phase at 30˚C. Y-axis shows 1,000*fold-change compared to the number of *ACT1* mRNA molecules (B) RT-qPCR measuring the level of the long and short mRNA isoforms of *PKP1* in a P$_{GAL7}$-*2xFlag-CBP-TUR1* strain in exponential phase and stationary phase at 30˚C, cultured in YPD (glucose) or YPGAL (galactose). Y-axis shows 1,000*fold-change compared to the number of *ACT1* mRNA molecules. Experiments were performed in biological triplicates and technical duplicates. Error bars are shown median +/− standard deviation. (C) Western blot analysis of *TUR1* conditional expression. Tur1 was detected using an anti-flag antibody and actin was used as control. The data underlying this figure can be found in S1 Data. TSS, transcription start site; WT, wild type.

genes with at least 2 TSSs and at least 20 TSS reads in all 4 samples (WT E30, WT S30, *tur1Δ* E30, and *tur1Δ* S30) (S9 Table). As expected, the stationary phase enhanced usage of the second TSS of *PKP1* in the WT strain, whereas usage of this distal TSS is nearly abolished in the *tur1Δ* background confirming the Tur1-dependent altTSS usage regulation at this locus (Fig 10A and 10B). Strikingly, 91% (*n* = 155) of the 171 genes with altTSS significantly regulated by

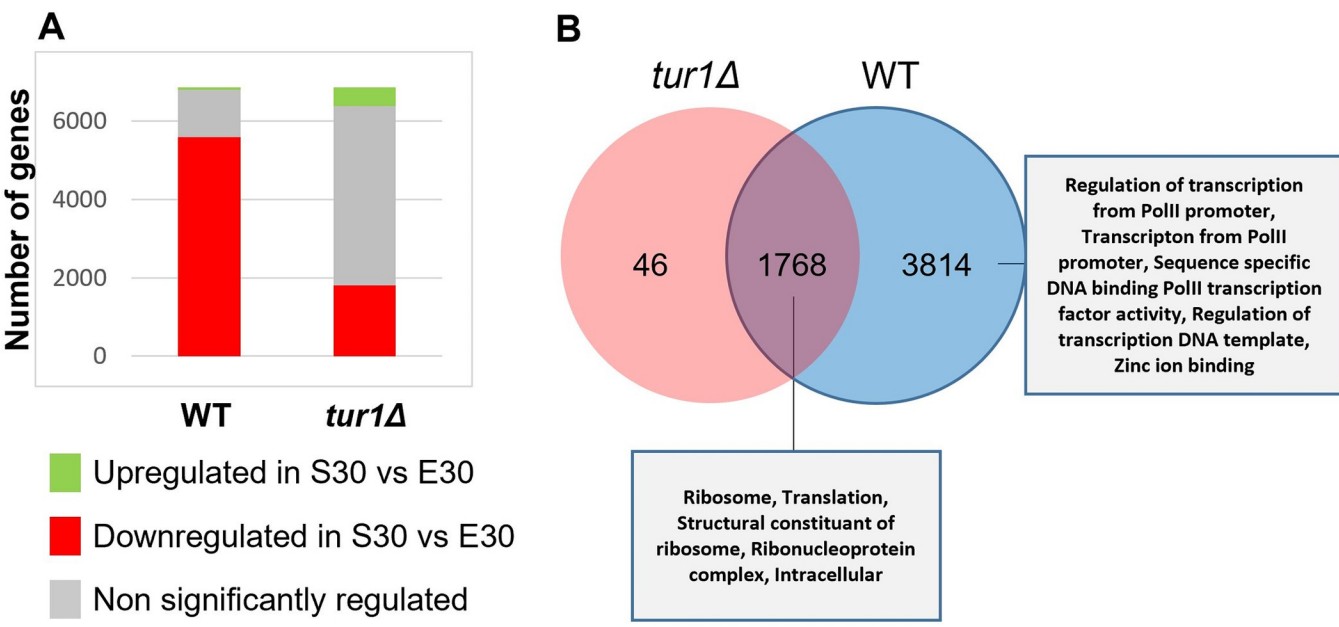

**Fig 9. Comparative differential gene expression analysis during the transition exponential to stationary phase in the WT and *tur1Δ* strains.** (A) Number of genes significantly regulated in WT and *tur1Δ* strains during the exponential to stationary phase transition at 30°C determined by DeSEQ2 (2-fold change, adjusted *p*-value <0.05). (B) Venn diagram describing the overlap of genes down-regulated during the exponential to stationary phase transition at 30°C in WT and *tur1Δ* strains. The data underlying this figure can be found in S1 Data. WT, wild type.

the growth phase in the WT were not regulated anymore in a *tur1Δ* background (Fig 10A). For instance, at the *IDP1* gene (CNAG_03920), which encodes a putative mitochondrial isocitrate dehydrogenase, cells growing in exponential phase mostly use the first TSS which promotes

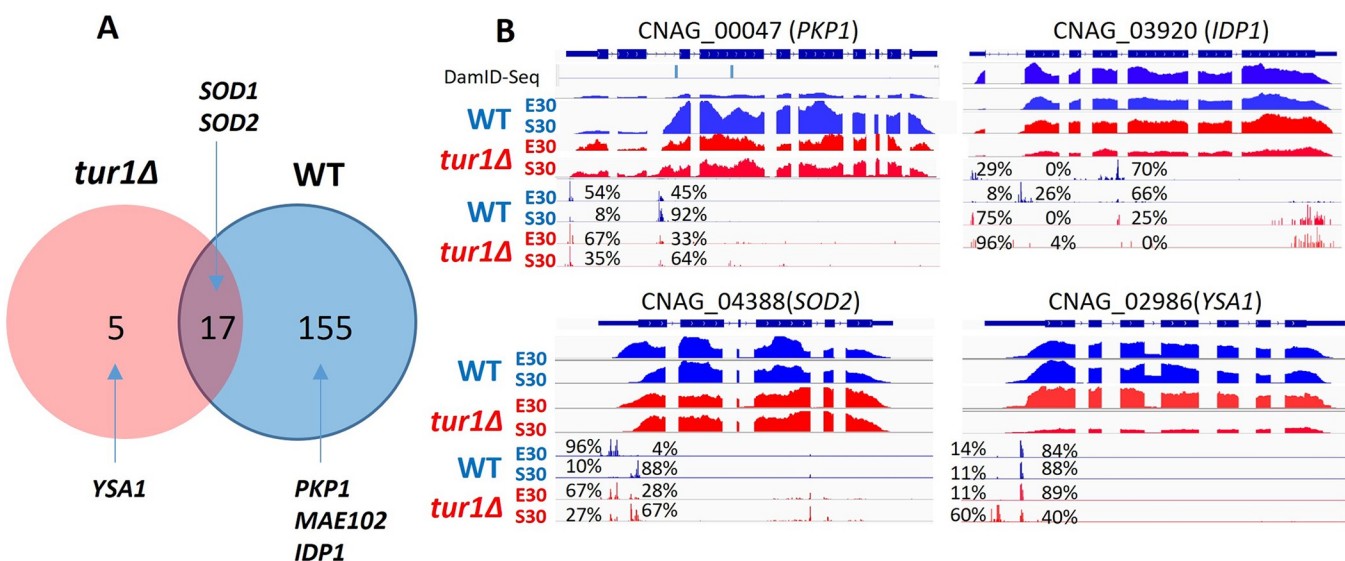

**Fig 10. Tur1 is a global regulator of altTSS usage in *C. neoformans*.** (A) Venn diagram depicting the overlap between altTSS regulated genes during transition from exponential phase to stationary phase in the WT and *tur1Δ* strains. (B) IGV visualization of RNA-seq and TSS-seq data at representative altTSS regulated genes in WT and *tur1Δ* strains grown in exponential phase (E30) and stationary phase (S30) at 30°C. The percentages indicate the proportion of each TSS cluster used in each condition. Detailed numbers are given in S9 Table. The position of the DamID-Seq methylated residues in exponential phase are indicated at the locus CNAG_00047. TSS, transcription start site; WT, wild type.

the production of a long mitochondrial isoform, whereas in stationary phase, the second TSS is mainly used promoting the synthesis of a short cytosolic isoform (Fig 10B). In the absence of *TUR1*, this growth phase-dependent regulation is lost, and the long isoform is expressed in both exponential and stationary phases (Fig 10B). In contrast, at some loci, deletion of *TUR1* has little impact on altTSS usage (Fig 10A). For instance, Tur1 has only a minor role if any on the altTSS usage regulation by the growth phase at the *SOD1* and *SOD2* genes (Fig 10B). *TUR1* deletion also revealed new growth phase-dependent altTSS usage regulation in *C. neoformans*. Thus, we identified 5 genes for which this regulation can be only observed in the *tur1Δ* context (Fig 10). The *YSA1* gene coding for an ADP-ribose pyrophosphatase is such an example. In that case, the long isoform produced in stationary phase in the *tur1Δ* strain contains an MTS absent in the current *C. neoformans* genome annotation (Fig 10B).

## A Tur1-dependent Mae102 subcellular targeting

As seen above, in a WT context, a growth phase-dependent altTSS usage regulation was observed at the *MAE102* gene. In that case, the second TSS is mostly used in exponential phase, whereas in stationary phase, the long isoform targeted to the mitochondria is expressed using the most upstream TSS (Fig 5A). *TUR1* deletion completely reversed this pattern (Fig 11A). Indeed, *tur1Δ* mutant grown in stationary phase preferentially expresses the short isoform, whereas in exponential phase it mostly uses the upstream TSS, only expressing the long isoform albeit at a low level (Fig 11B). To study the impact of *TUR1* deletion on Mae102 protein subcellular targeting, we deleted *TUR1* in a strain expressing a C-terminal mNeonGreen-tagged version of the Mae102 protein. In this genetic background, in exponential phase, Mae102 does not localize in the nucleus in agreement with the absence of production of the short isoform (Fig 5B). In stationary phase, under which the short isoform is strongly induced in *tur1Δ*, Mae102 seems to be poorly targeted to the mitochondria. It is noteworthy that Mae102 is not targeted to the nucleus either suggesting that additional factors specifically present in exponential phase are necessary for targeting this protein to the nucleus. Overall, this analysis exemplifies how Tur1 regulates altTSS usage to control protein subcellular targeting during the transition between exponential to stationary phase (Fig 5B).

## Tur1 could act directly on a subset of genes

To determine whether the effect of Tur1 on the regulation of aTSS usage during the transition from exponential to stationary phase could be direct, we constructed a strain expressing a 2× flag tagged version of Tur1 under the control of the actin gene promoter. After phenotypically validating the strain, we used it to perform ChIP-Seq analysis on chromatin purified from cells grown in exponential and stationary phases in triplicate. After peak calling using MASC2 [48], we considered positions where a peak was identified in all 3 stationary phase replicates (151 peaks), or all 3 exponential replicates (29 peaks), or both (400 peaks) to be robust Tur1 binding sites (S10 Table). These 580 Tur1 binding sites were associated with 713 genes. Nearly 60% ($n = 414$) of these genes were down-regulated in WT during the transition from exponential to stationary phase, but not in the *tur1Δ* strain, suggesting a direct regulation by Tur1 in this process. Finally, we found that 13% ($n = 20$) of the genes regulated in a Tur1-dependent manner by alternative TSSs during this phase transition were also bound by Tur1, suggesting a direct role of Tur1 in this regulation at these loci. However, the patterns of ChIP-seq read alignment were not easy to interpret and overall, these ChIP-seq experiments remained partly inconclusive regarding the mechanism by which Tur1 regulates alternative TSS usage in *Cryptococcus*. We reasoned that the use of a strong promoter to express the tagged version of Tur1 might have confounded our ChIP-Seq results by masking potential regulation. To overcome this

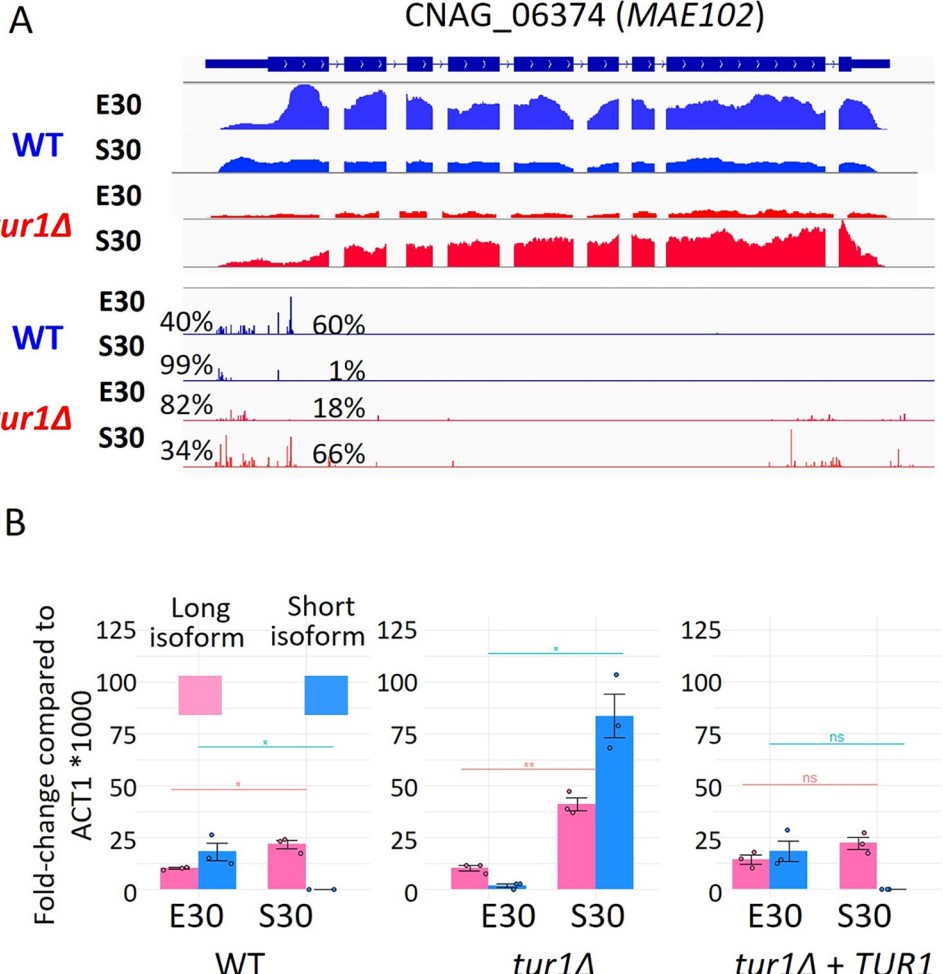

**Fig 11. Tur1 regulates altTSS usage at *MAE102*.** (A) IGV visualization of RNA-seq and TSS-seq of CNAG_06374 in WT and *tur1Δ* strain grown in exponential phase (E30) and stationary phase (S30) at 30°C. The percentages indicate the proportion of each TSS cluster used in each condition. Detailed numbers are given in S9 Table. (B) RT-qPCR analysis of the level of the long and short *MAE102* mRNA isoforms in WT, *tur1Δ*, and complemented *tur1Δ + SH1::TUR1* strains grown in exponential or stationary phase at 30°C. Y-axis shows 1,000*fold-change compared to the number of *ACT1* mRNA molecules. Experiments were performed in biological triplicates and technical duplicates. Error bars are shown median +/− standard deviation. The data underlying this figure can be found in S1 Data. TSS, transcription start site; WT, wild type.

problem, we constructed a strain expressing a Dam-Tur1 fusion protein under the control of the native *TUR1* promoter and performed DamID-Seq analysis [49] using DNA purified from cells grown in exponential and stationary phases in triplicate. Analysis of the data revealed 1,698 adenine residues associated with 1,283 genes that were specifically methylated in either exponential or stationary phase, suggesting genome-wide regulated Tur1 binding (S11 Table). As for our ChIP-Seq analysis, nearly 60% ($n$ = 731) of these genes were down-regulated in WT during the transition from exponential to stationary phase but not in the *tur1Δ* strain, suggesting direct regulation. However, the overlap between the 2 lists was only 13% ($n$ = 106). We found that 21% of the genes ($n$ = 32) regulated by aTSS in a Tur1-dependent manner contained methylated adenine residues. Interestingly, a highly regulated adenine residue was identified very close to the second TSS in *PKP1*, suggesting regulated Tur1 binding (Fig 10B).

Overall, both strategies indicate that Tur1 does indeed bind DNA of some genes regulated by alternative TSSs, suggesting that at least for some genes the regulation may be direct. However, the pattern of Tur1 binding regulation during the transition from exponential to stationary phase suggested by the 2 strategies did not provide a straightforward model for Tur1-dependent aTSS regulation.

## Tur1 regulates oxidative stress sensitivity and resistance to macrophage phagocytosis

Tur1 was previously reported to regulate mating efficiency, sensitivity to menadione and cadmium, as well as virulence in a mouse infection model [44]. We confirmed the sensitivity of *tur1Δ* mutant to menadione, a commonly used superoxide generator and, this sensitivity was reverted in the *tur1Δ+TUR1* complemented strain (Fig 12A). However, the *tur1Δ* mutant does not display growth defect in the presence of reactive oxygen species (ROS) such as $H_2O_2$ [44], 0.2 mM Cumen hydroperoxide, or 2 mM $NaNO_2$ (S7 Fig). This suggests the protective function of Tur1 against ROS generator is specific to superoxide generator menadione, consistent with the fact that cells employ distinct mechanisms to adapt to different ROS [50]. Similarly, we observed no growth defect of the *tur1Δ* strain in the presence of an inhibitor of superoxide dismutase (sodium dithiocarbamate (100 mM)) (S7 Fig). This result is in good agreement with the fact that Tur1 does not affect neither the expression of *SOD1* and *SOD2* nor the altTSS usage at these loci (Fig 10). STRING 1.5 [51] analysis of the genes regulated by altTSS in a Tur1-dependent manner showed an enrichment in GO-terms related to translation (large and small ribosome subunits assembly, translation initiation) and also genes related to glycolysis and glycogenesis (glycolytic and pyruvate metabolic processes). We also identified several mitochondrial protein encoding genes among those regulated by altTSS in a Tur1-dependent manner. We thus thought to test the sensitivity of *tur1Δ* to inhibitors of different components of the respiratory chain: 0.4 mM rotenone (complex I), 10 mM Dimethyl malonate (complex II), 5 μg/ml antimycin A (complex III), and 0.2 μg/ml oligomycin (complex V). We found that the growth of the *tur1Δ* mutant is severely impeded in the presence of 5 μg/ml antimycin A, a

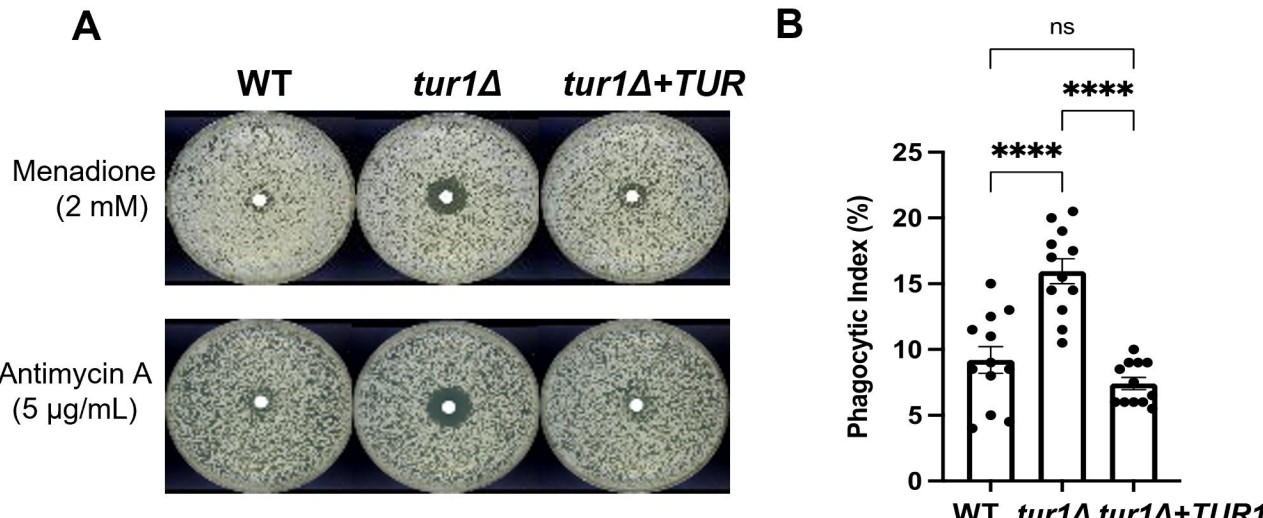

**Fig 12. Tur1 regulates resistance to menadione, antimycin A, and macrophage phagocytosis.** (A) Results of resistance disc assays; 2 mM menadione and 5 μg/ml antimycin A filter discs were used on YPD plates. Plates were incubated at 30˚C for 3 days and photographed. (B) Macrophage phagocytosis assay shows the PI for unopsonized *C. neoformans* WT, *tur1Δ*, and complemented *tur1Δ+TUR1* strains as described in Material and methods. The data underlying this figure can be found in S1 Data. PI, phagocytic index; WT, wild type.

known inducer of mitochondrial superoxide production [52], but not affected by the other agents (Figs 12A and S7). These results suggest that Tur1 might contribute to oxidative stress defense in *C. neoformans*. Interestingly, *tur1Δ* mutant cells are also phagocytosed significantly more efficiently than both the WT and the complemented strain (Fig 12B). Collectively, these results suggest that Tur1 might regulate *C. neoformans* virulence by somehow regulating its sensitivity to ROS and its resistance to phagocytosis.

## Discussion

There is no accepted standard bioinformatics pipeline to analyze TSS-seq data mostly because the structure and density of genes are very different between organisms and because the strategies used to produce this type of sequencing data are diverse [5,17,53–55]. Here, we report a *Cryptococcus*-adapted bioinformatics protocol to analyze TSS-seq data. We thus identified thousands of TSS clusters associated with coding genes in 2 species of pathogenic *Cryptococcus*. The structure of these TSS clusters (i.e., broad or sharp) and the preferential association of sharp TSS clusters with TATA-box containing promoters and highly regulated genes resemble results previously obtained in some other reference organisms [2,8,12,34,54,56–58]. This suggests that our strategy is robust and that the TSS clusters identified are genuine. Moreover, alternative TSS usage analysis identified thousands of alternative usage events which impact nearly a quarter of the coding genes in both *Cryptococcus* species. Because we have been very conservative in our bioinformatics analysis and because we only compared 4 conditions of growth for this study, our data suggest that altTSS usage might be far more widespread. In fact, altTSS usage is probably a major regulator of gene expression as well as transcriptome and proteome diversity in *Cryptococcus* as observed in metazoans [5,6,11,34] and for some aspects, in the 2 model yeasts [16,17,19].

Regulation by alternative TSS can impact gene expression by altering transcript stability as it has been previously shown for alternative splicing [27]. Although both regulations have the potential to greatly impact the transcriptome by inducing transcripts degradation, they differ in several aspects. First, altTSS-mediated regulation is predicted to mostly rely on cytoplasmic degradation by NMD, whereas the decay induced by intron-retention is NMD-independent and occurs in the nucleus [27]. Besides, the extent of the impact of these 2 means of regulation is dramatically different. Indeed, we previously showed that the impact of alternative splicing is probably minor, whereas the data we present in this study suggest a remarkable regulatory potential for altTSS-mediated regulation. First, as previously reported in baker's yeast [59,60] and more recently genome-wide in plants [61], number of altTSS clusters are positioned close to the annotated ATG in *Cryptococcus*. As shown, growth conditions can regulate altTSS usage at *MAE102* and promote the synthesis of protein isoforms with different N-terminal sequences and different subcellular targeting. Second, in both *Cryptococcus* species, we identified close to 2,000 altTSSs positioned within the coding sequence of genes and promoting TRASS transcription. These internal TSS have been previously reported in metazoans [62] and a few examples exist in model yeasts in which their usage can be also regulated by the growth conditions and/or the genetic background [16,19,22,63,64]. For example, in *S. cerevisiae* 2 studies identified a few dozen genes where internal TSS usage is regulated by cell-fate transition during gametogenesis [19,64]. However, the coding potential of the resulting transcripts has been difficult to evaluate. Since most peptides potentially encoded by these truncated transcripts, including the ones we identified in this study, are in frame with the full-length protein, classical proteomics is not able to specifically identify them. Nevertheless, indirect evidence for the existence of these truncated proteins has been presented. For instance, in *S. cerevisiae*, cell-fate transition-specific truncated transcripts have been shown to overlap with ribosome protected

fragments [19] and in *S. pombe*, "disruptive" internal altTSS located within or downstream of conserved protein domains are generally poorly used. Yet, only a few examples of functional truncated proteins have been reported [62,64,65]. Here, N-terminomic approach strongly suggests that a significant part of these transcripts is translated. Our results are in good agreement with *S. cerevisiae* data showing that some "internal cryptic" transcripts exhibiting 5′ truncation can be translated from downstream in-frame start codons and thus giving rise to shorter polypeptides [22]. As for baker's yeast, the function of these TRASS translated peptides is not known. Similarly, the function of all the other TRASS which might not be translated is unknown, though we can speculate that in some cases at least, the usage of one TSS might impact nucleosome positioning and chromatin modifications to regulate positively or negatively the transcription of the other TSS as shown in yeast [19,66,67]. In some other cases, the TRASS might have a function in trans which remains to be discovered.

Our analysis reveals an extended regulation of altTSS usage in response to change in growth conditions in *Cryptococcus*. In that sense, these results are similar to results obtained in *S. pombe* in which 42% of the genes are regulated by altTSS usage in response to environmental cues [16]. The study of *S. pombe* alternative TSS usage identified specific motifs associated with altTSS according to the type of stress or the condition of growth tested [16] suggesting specific regulation of altTSS by TFs. This is in good agreement with several other studies in baker's yeast. For instance, Ndt80 regulates the expression of internal transcripts during gametogenesis [19,64], whereas Ume6 controls the expression of long undecoded transcript isoforms (LUTIs) during meiosis [66]. In Valine and Isoleucine restricted conditions, Gcn4 can bind within the CDS or the 3′ UTR of about 300 genes where it can promote bidirectional transcription [63]. These data suggest the existence of a programmed regulation of altTSS usage depending on the growth condition in addition to the classically identified regulons. Similarly in plants, light activates phytochromes which will act on TF activities to coordinate altTSS usage impacting both the uORF-mediated inhibition of gene expression and protein localization [61,68]. In humans, a specific pattern of altTSS usage has been reported according to cell type, developmental stage, or cancer [69,70]. Accordingly, we here identified a basidiomycete-specific TF regulating altTSS usage during the transition from exponential phase to stationary phase in *C. neoformans*. In the absence of this TF, the altTSS usage during this change in cell physiology is globally impaired. In pathogenic and nonpathogenic yeasts, entry to stationary phase is usually characterized by a slowdown of cell metabolism through the action of several TFs [71–73]. It is predictable that such program alteration of altTSS usage exists in all these fungi tuning the transcriptome and cell metabolism toward a quiescent state.

Our analysis suggests that Tur1 is a global regulator of aTSS usage in *C. neoformans* during the transition from exponential to stationary phase. Our ChIP-seq and DamId-seq analysis suggests both a direct and indirect mode of action depending on the locus. Interestingly, our data suggest that Tur1 binds to the internal sequence of *PKP1* only during exponential phase, although its absence does not affect alternative TSS usage during this phase. This effect of *TUR1* deletion is restricted to the stationary phase, during which it is required to regulate TSS2 usage but does not bind *PKP1*. These results suggest a model (Fig 13) in which Tur1 binding during exponential phase is required for the recruitment/activation of another TF that will bind to the Tur1 internal promoter during stationary phase. This change in TF usage could be due to a change in chromatin structure due to the increase in TSS1 usage observed during this phase of growth. Indeed, modification of the transcriptional sense or antisense can alter the chromatin structure allowing TF binding as previously shown [18,74]. It is important to say that the present model is probably an oversimplification of what happens during the 10 h between the exponential and stationary time points during which the switch is TSS usage can be observed. However, this type of TF-TF interaction associated with TF exchange and

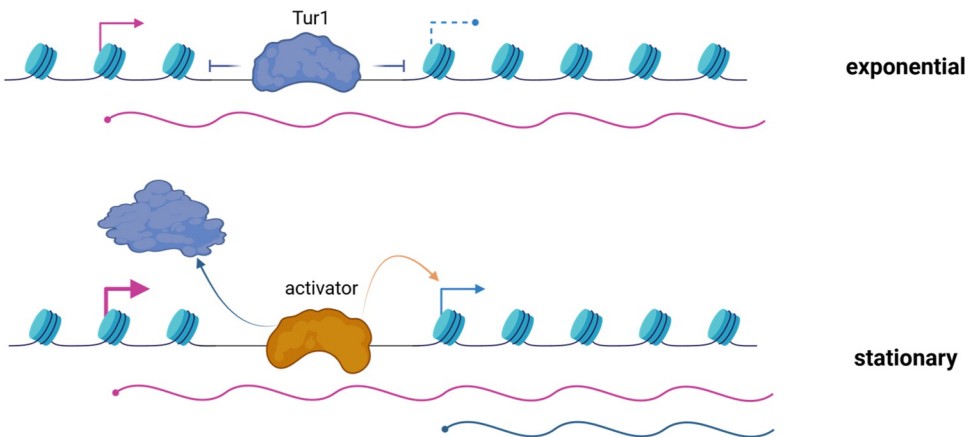

**Fig 13. Tentative model for Tur1 start site selection.** At the *PKP1* locus, Tur1 binds to an open chromatin region during exponential phase. In stationary phase, the increased usage of TSS1 changes the chromatin structure and Tur1 may recruit another TF to activate TSS2 usage. In addition, Tur1 could control the expression of positive or negative regulators acting at other sites regulated by aTSS. TF, transcription factor; TSS, transcription start site.

alternative transcript formation has been described in the literature. For example, in fission yeast during glucose starvation, orchestrated exchanges and interactions between the TFs Tup11/12, Rst2, Atf1, and Php5 resulted in the transition from a stage where only a long RNA (mlonRNA) is transcribed to a stage where the short functional Flp1 mRNA is produced [75]. Finally, since we also observed Tur1-dependent aTSS regulation at loci where we did not observe Tur1 binding, we must also consider Tur1 as an indirect regulator of aTSS usage, controlling the expression of unknown direct positive or negative regulators. Accordingly, 48 TFs are regulated in a Tur1-dependent manner during the transition from exponential to stationary phase.

Tur1 regulates virulence in *C. neoformans* but does not regulate the expression of its main virulence factors (i.e., capsule biosynthesis, melanin production, or fitness at 30˚C or 37˚C). However, *TUR1* deletion impacts resistance to superoxide stress as well as the phagocytosis of *C. neoformans* by macrophages, phenotypes previously shown to impact virulence in *C. neoformans* [76,77]. More specifically, antimycin A treatment is known to drastically induce extra-mitochondrial superoxide release [78]. Given that Tur1 regulates the localization of mitochondrial enzymes like Mae102, it could potentially contribute to mitochondrial metabolism's defense against oxidative stress by facilitating the distribution of these enzymes. However, further experiments are necessary to establish a conclusive link and determine a direct impact.

Besides the model yeasts and *Cryptococcus*, altTSS has been poorly analyzed in fungi [18]. Nevertheless, specific examples in *Neurospora crassa*, *Aspergillus oryzae*, *Fusarium graminearum*, and *Candida glabrata* suggest a widespread although completely ignored regulation of the altTSS usage in pathogenic fungi [31,79–81]. We identified thousands of new RNA molecules resulting from the usage of altTSS and showed that at least a subset of them are coding for new proteins or proteins isoforms with a different subcellular localization in *Cryptococcus*. We can anticipate that the same will be true for all other fungal pathogens. Our work suggests that the impact of altTSS usage in their biology and virulence might be major. This study opens up new possibilities for exploring the interactions between host pathogens and emerging players, as well as uncovering insights into gene regulation and the structure of the transcriptome in fungi.

## Material and methods

### DNA and RNA sequencing

Each *Cryptococcus* cell preparation was spiked in with one tenth (OD/OD) of *S. cerevisiae* strain FY834 [82] and cells were grown in YPD at 30˚C in exponential phase. Cells were washed, snap frozen, and used to prepare RNA and total DNA samples as previously described [83,84]. For RNA-seq, strand-specific, paired-end cDNA libraries were prepared from 10 μg of total RNA by polyA selection using the TruSeq Stranded mRNA kit (Illumina) according to manufacturer's instructions. cDNA fragments of approximately 400 bp were purified from each library and confirmed for quality by Bioanalyzer (Agilent). DNA-seq libraries were prepared using the kit TruSeq DNA PCR-Free (Illumina). Then, 100 bases were sequenced from both ends using an Illumina HiSeq2500 instrument according to the manufacturer's instructions (Illumina). Trimmed reads were mapped to the *Cryptococcus* and *S. cerevisiae* genomes using Bowtie2 [85] and Tophat2 [86] as previously described [14]. RNA-seq reads counts were then normalized considering the ratios of DNA and RNA aligned to each genomes using the strategy previously described by Malabat and colleagues [53]. Differential gene expression analysis was performed using the DESeq2 package [87] using default settings but ommiting the associated read count normalization step.

TSS-seq libraries preparations were performed as previously described [14,53] replacing the TAP enzyme by the Cap-Clip Acid Pyrophosphatase (Cellscript). Sequencing was performed on Illumina HiSeq2500 to obtain 50 base single end reads. Data processing including reads trimming, clipping, and mapping were performed as previously described [14]. TSS reads were first clustered using arbitrarily chosen maximum intra-cluster distance (d = 50 nt) between sites. We then looked for the optimal "d" value to define clusters, reasoning that an excessively small value would separate reads belonging to a single cluster into several ones, whereas an overestimated "d" would merge reads from different clusters into a single one. We thus tested increasing values of "d" (from d = 1 to d = 50) reasoning that the size of the clusters thus defined will also increase until their actual boundaries would be reached. We here considered only the genes with only 1 TSS cluster in their TL sequence using the exponential 30˚C dataset and d = 50 defined. We then plotted the "d" values with the percentage "P" of clusters reaching their maximum cluster size (S8A Fig). We then used the nls function in R to modelized this relationship in the following equation where $P = 100/(1+exp(-0.1655*d)^{4.2957})$ (S8B Fig). With the model, the correlation of d and P reached its peak at d = 17 (S8B Fig). We thus choose this value for our analysis.

### ChIP-Seq analysis

ChIP-Seq analysis were performed as previously described [88]. Briefly, cells were grown in 100 ml YPD medium at 30˚C to a final A600 nm at 0.8 to 1.0 (exponential phase) or for 16 h (stationary phase). Cells were then crosslinked by addition of 2.7 ml of 37% formaldehyde (1% final concentration) and let for 10 min at 25˚C with gentle shaking (70 rpm). Crosslinking reaction was stopped by addition of 8 ml of 2.5 M Glycine and let for 10 min at 25˚C with gentle shaking (70 rpm). Cells were harvested by 5 min centrifugation at 3,000 ×g, washed with PBS and pellets from half of the culture were resuspended in 500 μl lysis buffer (50 mM Hepes-KOH at pH 7.5, 140 mM NaCl, 1 mM EDTA, 1% Triton-X100, 0.1% sodium deoxycholate, 1 mM PMSF, and 1X EDTA-free Complete protease inhibitor cocktail (Roche)). Approximately 500 μl of 0.5 mm diameter glass beads were added and cells were lysed by 8 runs of 40 s of bead beating (6.5 m/s, FastPrep-24, MP Biomedicals) with cooling on ice between each round. Cell lysate was collected by centrifugation at 2,000 ×g for 1 min

after piercing the bottom of the tube with a needle and placing it on top of a clean tube. This step was repeated after addition of 500 μl of lysis buffer on the glass beads. Chromatin was fragmented using a Bioruptor Pico (diagenode) for 15 cycles (30 s on, 30 s off) to give fragments size between 200 bp and 500 bp. Samples were centrifuged at 16,000 ×g for 10 min and supernatants were recovered, hereafter referred to as chromatin extracts, and 20 μl of each chromatin extract were kept aside for "input" samples. Immunoprecipitation. Remaining chromatin extracts were incubated overnight at 4˚C with 50 μl of prewashed anti-Flag M2 magnetic beads suspension (Millipore, Merck M8823) per sample. The day after, beads were washed twice with 1 ml of lysis buffer, twice with 1 ml "high salt" lysis buffer (as lysis buffer except for NaCl which is increased up to 360 mM), once with wash buffer (10 mM Tris-HCl (pH 8.0), 250 mM LiCl, 0.5% NP-40, 0.5% sodium deoxycholate, 1 mM EDTA) and once with 1 ml TE. ChIP were eluted twice for 10 min at 80˚C in 100 μl TE-SDS (10 mM Tris-HCl (pH 8.0), 1 mM EDTA, 1% SDS). DNA recovery. Decrosslink was achieved by adding 15 μl of 20 mg/ml proteinase K to eluates and overnight incubation at 65˚C. The day after, DNA was recovered using phenol–chloroform method, quantified by QuBit and sent for sequencingfor Illumina sequencing to Novogen (UK). After trimming, the 150 paired ends reads were mapped using Minimap2 [89]. Peak calling was performed using MACS2 [48] using default settings.

## DamID-Seq analysis

DamID was essentially performed as described in [49]. Briefly, for DpnII digestion and alkaline phosphatase treatment, 2.5 μg of genomic DNA extracted using the phenol–chloroform method and treated with RNase A were digested with 10 units of DpnII (NEB, R0543S) in a final volume of 20 μl at 37˚C for 6 h. Enzyme was inactivated 20 min at 65˚C. Samples were then treated with 5 units of Antartic phosphatase (NEB M0289S) in a final volume of 50 μl for 1 h at 37˚C. Phosphatase was inactivated 10 min at 70˚C and reaction was cleaned up using nucleospin PCR clean-up XS column (Macherey Nagel, 740611) with elution in 12 μl of elution buffer. For DpnI digestion, 5 μl of the eluate was digested overnight at 37˚C with 10 units of DpnI (Takara, 1235A) in a final volume of 10 μl. Enzyme was inactivated 20 min at 80˚C. Ligation of the pre-annealed dsAdR (AdRt/AdRb) adapter duplex to DNA fragments was performed overnight at 16˚C in a final volume of 20 μl with 2.5 μm dsAdR and 400 units of T4 DNA ligase (NEB M0202S). Reaction was cleaned up using nucleospin PCR clean-up XS column (Macherey Nagel, 740611) and eluted in 50 μl of elution buffer. PCR with Advantage 2 (Takara 639207); 5 μl of ligation sample, 2.5 μl of 10X Advantage Buffer, 0.5 μl of 10 mM dNTP, 0.5 μl 10 μm AdR_PCR primer, and 0.5 μl of 50X polymerase mix were mixed in a final volume of 25 μl. PCR was carried out as follows: 68˚C for 10 min; 1 cycle of 94˚C for 3 min, 65˚C for 5 min, and 68˚C for 15 min; 4 cycles of 94˚C for 1 min, 65˚C for 1 min, and 68˚C for 10 min; and 17 cycles of 94˚C for 1 min, 65˚C for 1 min, and 68˚C for 2 min. Eight PCR reaction of 25 μl were performed for each sample, pooled and cleaned up together before sending for Illumina sequencing (Novogene). After triming of illumina adaptors, the DamID adaptor (GGTCGCGGCCGAG+GATC) containing reads were selected and aligned to the *C. neoformans* reference genome using Minimap2 [89]. Adenine residues of genomic GATC sequences positionned at the extremities of these reads were considered as potential methylated positions. The number of reads were normalized by the total number of reads aligned to the genome in each samples. To identify adenine methylated residues, we substracted the average number of reads obtained in the 3 control samples to the number reads obtained in the experimental samples in same condition. Negative numbers were considered as zeros.

### Calculating thermodynamic entropy-related shape index (SI) of TSS cluster

The SI index was calculated as previously demonstrated by Hoskin and colleagues [34] as below:

$$SI = 2 + \sum_{i}^{L} p_i \log_2 p_i,$$

where p is the probability of observing a TSS at base position i within the cluster, and L is the set of positions that possess mapped reads in the cluster.

### TSS cluster subpopulation detection and cluster shape classification

We considered only genes with only 1 TSS cluster in their TL and kept only clusters with both size and SI between 2.5% and 97.5% quantiles, as evaluated using the data obtained in exponential phase 30˚C (719 genes in exponential phase 30˚C, 691 in stationary phase 30˚C, 635 genes in exponential 37˚C, 586 genes in stationary phase 37˚C). To detect if there is more than 1 population of TSS clusters, the first principal component (PC1) of size and SI was used as input for the function "emtest.norm" of R package "'MixtureInf' [35] with the default value of the finite mixture model under null hypothesis m0 = 1. The output of function "emtest.norm" includes the percentage of each subpopulation together with the detected models with theirs mean and standard deviation. This information was then supplied to the argument "theta" of the function "plotmix.norm" to plot histogram of the observations and the fitted density of a mixture of models. TSS clusters were then classified into 2 groups based on the mean value of PC1 plus and minus 1 standard deviation of each group as returned by the function "emtest. norm."

### Motif enrichment discovery in core promoter region

To find motifs enriched in promoter regions, the genomic sequences from −100 to +30 of the major TSS peaks were extracted and used as input in MEME Suite software 5.4.1 [36] (de novo pattern discovery mode). Default settings were applied except these following parameters: number of motifs: 6, minimum width: 6, maximum width: 10, and search given strand only. The 6 motifs found were then scanned in each group of Sharp and Broad clusters in each condition with Fimo mode (threshold $p$-value $\leq 0.0005$) to examine their number and positions in each dataset.

### altTSS usage analysis

The "expression level" of each TSS cluster was calculated for each replicate and normalized by the total number of reads associated with gene containing the cluster. We did not consider the clusters that never contribute more than 10% total reads associated with a given gene in any replicate. We then performed 4 comparisons: EXPO 30 versus STAT 30, EXPO 37 versus STAT 37, EXPO 30 versus EXPO 37, and STAT 30 versus STAT 37.

### N-terminomic experiments

**Lysis and protein extraction.** Cells were lysed by sonication in lysis buffer containing 0.5% SDS, 100 mM HEPES (pH 8), 10 mM DTT, 10 mM EDTA, supplemented with protease inhibitors (Complete Mini EDTA-free, Roche). After sonication, the lysate was incubated for 20 min at RT with 1 µl of benzoase and centrifuged for 10 min at 12,000 $g$ at 4˚C. The supernatant was reduced and alkylated using 20 mM TCEP and 10 mM NEM (N-ethylmaleimide) respectively, for 30 min at RT in the dark. Protein concentration was determined with DC assay (BioRad) and 200 µg of each sample were further processed.

**TAILS protocol (terminal amine isotopic labeling of substrates).** TAILS experiment was based on reference protocol [42] with some modifications. Prior to labeling, the samples were first precipitated using acetone–methanol method to remove all primary amines. Briefly, 8 sample volumes of ice-cold acetone and 1 volume of ice-cold methanol were added, samples were let to precipitate overnight at −80˚C, centrifuged for 30 min at 14,000 *g* at 4˚C, and the pellet washed twice with ice-cold methanol. The dried pellet was resuspended in 1% SDS, 100 mM TEAB (pH 8) and incubated for 5 min at 95˚C. pH was adjusted to 7.5. Dimethylation labeling was carried out by adding 40 mM formaldehyde and 20 mM sodium cyanoborohydride ($NaBH_3CN$) and incubating overnight at 37˚C. The reaction was quenched by addition of 500 mM of Tris-HCl (pH 6.8) and incubation for 1 h at 37˚C.

**Tryptic digestion.** S-Trap mini spin column (Protifi, Hutington, United States of America) digestion was performed on dimethylated samples, according to manufacturer's instructions. Briefly, SDS concentration was first adjusted to 5% and aqueous phosphoric acid was then added to a final concentration of 2.5% following by the addition of S-Trap binding buffer (90% aqueous methanol, 100 mM TEAB (pH 7.1)). Mixtures were loaded on S-Trap columns by 30 s centrifugation at 4,000 *g*. Six washes were performed before adding the trypsin (Promega) at 1/20 ratio and incubation for 2 h at 47˚C. After elution, 10 μg of peptides were kept for MS/MS analysis and the remaining samples were vacuum dried.

**Negative selection by HPG-ALDII polymer.** Dried peptides were resuspended in 100 μl of 100 mM HEPES (pH 7). The HPG-ALDII polymer was added to samples at a polymer/peptide ratio 5:1 (w/w), followed by addition of 20 mM $NaBH_3CN$. Samples were then incubated overnight at 37˚C, and 100 mM Tris-HCl (pH 7) was added and samples were incubated for 30 min at 37˚C before loading by centrifuging for 10 min at 12,000 *g* on 10 kDa Amicon (Millipore) centrifugation device previously washed with 100 mM NaOH and water. The flow-through was collected and additional washing step was performed. The filters were then rinsed with water, polymer discarded and filters spined-down in upside-down position for 3 min at 5,000 *g*. Collected fractions were pooled with flowthroughs and vacuum-dried.

**MS/MS analysis.** Peptides were resuspended in 10% ACN, 0.1% TFA in HPLC-grade water prior to MS analysis. For each run, 1/10 of peptide solution was injected in a nanoRSLC-Q Exactive PLUS (RSLC Ultimate 3000) (Thermo Scientific, Waltham, Massachusetts, USA). Peptides were loaded onto a μ-precolumn (Acclaim PepMap 100 C18, cartridge, 300 μm i.d. × 5 mm, 5 μm) (Thermo Scientific) and were separated on a 50-cm reversed-phase liquid chromatographic column (0.075 mm ID, Acclaim PepMap 100, C18, 2 μm) (Thermo Scientific). Chromatography solvents were (A) 0.1% formic acid in water; and (B) 80% acetonitrile, 0.08% formic acid. Peptides were eluted from the column with the following gradient 5% to 40% B (120 min), 40% to 80% (1 min). At 121 min, the gradient stayed at 80% for 5 min and, at 126 min, it returned to 5% to re-equilibrate the column for 20 min before the next injection. One blank was run between each replicate to prevent sample carryover. Peptides eluting from the column were analyzed by data-dependent MS/MS using top-10 acquisition method. Peptides were fragmented using higher-energy collisional dissociation (HCD). Briefly, the instrument settings were as follows: resolution was set to 70,000 for MS scans and 17,500 for the data-dependent MS/MS scans to increase speed. The MS AGC target was set to 3.106 counts with maximum injection time set to 200 ms, while MS/MS AGC target was set to 1.105 with maximum injection time set to 120 ms. The MS scan range was from 400 to 2,000 m/z. Dynamic exclusion was set to 30 s duration.

**Data analysis.** The MS files were processed with the MaxQuant software version 1.5.8.3 and searched with Andromeda search engine against the annotated *C. neofomans* database [14] but also against a database containing all alternative proteins potentially resulting for alternative TSS usage. To search parent mass and fragment ions, we set a mass deviation of 3 ppm and 20 ppm, respectively. The minimum peptide length was set to 7 amino acids and

strict specificity for trypsin cleavage was required, allowing up to 2 missed cleavage sites. NEM alkylation (Cys) was set as fixed modification, whereas oxidation (Met) and N-term acetylation, N-term Dimethylation (any N-term), Dimethylation (Lys) were set as variable modifications. The false discovery rates (FDRs) at the protein and peptide level were set to 1%. Scores were calculated in MaxQuant as described previously [90]. The reverse and common contaminants hits were removed from MaxQuant output. The output files "peptides" and "modificationSpecificPeptides" were combined and further analyzed on excel.

## Transcription factor mutant library screening

Total RNA (5 μg) extracted from each of the *C. neoformans* 150 TF mutant strains [44] was subjected to DNAse I treatment (Roche) to eliminate contaminating genomic DNA. A total of 1 μg of the DNase I-treated RNA was then used for reverse-transcription (RT) with Maxima First Strand cDNA Synthesis Kit (Thermo Scientific). qPCR experiments using CFX96 Touch Real-Time PCR Detection System (Bio-Rad) with SsoAdvanced Universal SYBR Green Supermix (Bio-Rad) according to manufacturer's instructions. The primer pairs used were specific of either long (in the region located between TSS1 and TSS2) or both isoforms (downstream of TSS2) of *PKP1* or *ACT1* (S1 Table). The number of short isoforms was calculated as the number of both *PKP1* RNA isoforms minus the number of long isoforms. The levels of the *PKP1* long and short isoforms were compared between WT and mutant strains. The level of expression of *ACT1* was used as a control.

## Construction of strains deleted and complemented for *TUR1* gene in *C. neoformans*

The *TUR1* gene was deleted in *C. neoformans* KN99α reference strain [26] (S2 Table) by inserting a NEO$^R$ cassette in place of the coding sequence by electroporation using a transient CRISPR-Cas9 expression (TRACE) system [91]. Plasmids and primers to amplify Cas9 and gRNA against *TUR1* are listed in S3 and S1 Tables, respectively. For the complemented strain, we PCR amplified a DNA fragment spanning a genomic DNA region form 1 kb upstream of annotated ATG to 1 kb downstream of stop codon of the gene *TUR1*. This fragment was then cloned into *pSDMA57* (S2 Table) allowing the integration at the genomic "safe haven 1" locus and complementation of *TUR1* in *tur1Δ::NAT* mutant [92]. For galactose-inducible complementation, the *TUR1* sequence from annotated ATG to 1 kb downstream of stop codon was amplified and then was fused at the 5′ end with 2xFLAG-CBP [93] amplified from plasmid pCM189-NTAP [94]. This construct was positioned under the control of the *GAL7* promoter in the plasmid pSDMA57-GAL7p-TUR1. The construct was then PCR-amplified and co-transformed into *tur1Δ* mutant strain together with a PCR-amplified DNA fragments containing the sequence of Cas9 and a PCR-amplified DNA fragment coding the gRNA targeting Safe Haven 1 as previously described [91]. For the construction of the $P_{ACT1}$::*2xFLAG-CBP*::*TUR1* strain, the sequence of the *ACT1* promoter was PCR amplified and used with the *2xFLAG-CBP*::*TUR1* strain before and being targeted to the Safe Haven 1 as previously described [91]. The resulting strain NE1605 was then used for ChiP-Seq analysis. Finally, a strain expressing the fusion protein Dam:Tur1 under the control of the *TUR1* native promoter was constructed using a similar strategy. Briefly, the *E. coli* DNA adenine methyltransferase gene was amplified from the plasmid kindly given by Eugene Gladyshev (Institut Pasteur) and fused with *TUR1* gene. The fusion PCR fragment was then targeted to the *TUR1* locus native locus after co-transformation of the *C. neoformans* Cas9 expressing strain CM2049 [95] with a *TUR1*-specific gRNA and a linear plasmid containing a nourseothricin resistance marker. The resulting strain NE1713 was used for DamID-Seq experiments.

## Construction of mNeon tagged *MAE102* genes, fused ATG, and mutated aATG mutant strains

For *MAE102* gene tagging, the region covering mNeonGreen-CBP-2xFlag-terminator together with NAT1 cassette was PCR amplified from pBHM2404 and fused in 5′ with 1 kb upstream the STOP codon of *MAE102* (without including it) and in 3′ with 1 kb downstream of the STOP codon to allow recombination in the genome (see S1 Table for primers). This cassette was co-transformed with a PCR fragment allowing production of a gRNA directed against the 3′ of *MAE102* in the strain CM2049 in which the Cas9 gene is integrated at the Safe Heaven 2 [95,96] and constitutively expressed. The *mae102-MTSΔ* mutant in which the region containing a putative MTS has been deleted was constructed using a similar strategy. Here, a cassette in which 1 kb upstream of the annotated ATG was fused with 1 kb starting at the second ATG was first constructed before being co-transformed with the NEO marker (pSDMA57 cut by PacI), these 2 constructs being targeted by appropriate gRNAs at the aATG of *MAE102* and at the Safe Haven 1 locus, respectively. These mutations were done on the mNeonGreen-tagged version of *MAE102*. Finally, the *mae102-M1R* mutant was constructed with the same strategy using a cassette in which the annotated ATG was replaced by a CGT. Primers, strains, and plasmids used to construct these mutants are listed in the S1, S2 and S3 Tables, respectively.

## Western blot

The *C. neoformans tur1Δ* mutant strain expressing the fusion protein 2xFlag-CBP-Tur1 under the control of the *GAL7* promoter [46] was cultured at 30˚C in media containing glucose (YPD) or galactose (YPGAL) until exponential phase or stationary phase. Total proteins were extracted from 10 $OD_{600}$ units of cells and were dissolved in 250 μl of 1X SDS-PAGE sample buffer, and 10 μl of protein extract were loaded on a denaturing polyacrylamide gel before being transferred to a nylon membrane and probed with an anti-flag M2 antibody (Sigma Aldrich F1804). An anti-β-actin antibody (Sigma Aldrich A2066) was used as control.

## Northern blot

RNA was extracted with TRIZOL Reagent (Thermo Fisher Scientific) following the manufacturer's instructions. Total RNA (10 μg) was separated by denaturing agarose gel electrophoresis and transferred onto Hybon-N+ membrane (Sigma-Aldrich) and probed with [$^{32}$P]dCTP-radiolabelled DNA fragments. The banding pattern was quantified with a Typhoon 9200 imager and Image Quantifier 5.2 sofware (Molecular Dynamics).

## Phenotypic assays

For spot assays, cells grown in liquid YPD medium at 30˚C for 16 h were counted, serially diluted, and spotted on solid YPD medium supplemented with chemical compounds purchased from Sigma Aldrich: 0.2 mM cumen hydroperoxide, 2 mM $NaNO_2$, 100 mM sodium dithiocarbamate, 0.4 mM rotenone, 10 mM dimethyl malonate, and 0.2 μg/ml oligomycin. Cells were incubated at 30˚C for 3 days and the plates were photographed.

For disc inhibition assays, overnight cultures of indicated *C. neoformans* strains were grown overnight. The following day cells were diluted to a final $OD_{600}$ of 0.1 and 100 μl was then spread on YPD plates. Filter discs containing 20 μl of 10 mM menadione or 5 μg/ml antimycin A were positioned at the center of the Petri dish and plates were incubated at 30˚C for 3 days and photographed.

## Phagocytosis assay

The murine macrophage J774A.1 cell line [97] was cultured in T-75 flasks (Fisher Scientific) in Dulbecco's modified Eagle medium (DMEM), low glucose (Sigma-Aldrich), containing 10% live fetal bovine serum (FBS) (Sigma-Aldrich), 2 mM L-glutamine (Sigma-Aldrich), and 1% Penicillin and Streptomycin solution (Sigma-Aldrich) at 37°C and 5% $CO_2$. Cells were passaged when 70% to 90% confluent by scraping and resuspending in fresh complete DMEM at a ratio of 1:3 to 1:6.

Phagocytosis assays were performed to measure the capacity of J774 macrophages to engulf *C. neoformans* WT, *tur1Δ*, and *tur1Δ+TUR1* strains. Twenty-four hours before the start of the phagocytosis assay, $1 \times 10^5$ J774 macrophages were seeded onto wells of a 24-well plate (Greiner Bio-One). The cells were then incubated overnight at 37°C and 5% $CO_2$. At the same time, a colony of each strain was picked from the stock plate and resuspended in 3 ml liquid YPD broth. The culture was incubated at 25°C overnight under constant rotation (20 rpm). On the day of the assay, macrophages were activated using 150 ng/ml phorbol 12-myristate 13-acetate (PMA) (Sigma-Aldrich) for 1 h at 37°C. PMA stimulation was performed in serum-free media to eliminate the contribution of complement proteins during phagocytosis. To prepare *C. neoformans* for infection, an overnight *C. neoformans* culture was washed 2 times in 1× PBS and centrifuged at 6,500 rpm for 2.5 min. To infect macrophages with unopsonised *C. neoformans*, after the final wash, the *C. neoformans* pellet was resuspended in 1 ml PBS, counted using a hematocytometer, and fungi incubated with macrophages at a multiplicity of infection (MOI) of 10:1. The infection was allowed to take place for 2 h at 37°C and 5% $CO_2$. Afterwards, extracellular *Cryptococcus* was washed off and wells were treated with 10 μg/ml calcofluor white (CFW) (Sigma-Aldrich) for 10 min at 37°C to distinguish between phagocytosed and extracellular *C. neoformans*. Images were acquired using the Nikon Eclipse Ti inverted microscope (Nikon) fitted with the QICAM Fast 1394 camera (Hamamatsu) under 20× magnification and using the phase contrast objective and CFW channel. Images were analyzed using the Fiji image processing software (ImageJ). Two hundred macrophages were counted and phagocytic index (PI) was quantified as the percentage of macrophages that phagocytosed at least 1 *Cryptococcus* cell (PI = ((number of infected macrophage/total number of macrophages counted) * 100%)).

## Supporting information

**S1 Table. List of primers used in this study.**
(XLSX)

**S2 Table. List of strains used in this study.**
(XLSX)

**S3 Table. List of plasmids used in this study.**
(XLSX)

**S4 Table. List of genes with altTSS clusters identified in *C. neoformans* and *C. deneoformans*.** GeneID of genes in *C. neoformans* or *C. deneoformans* and the ortholog genes in the other species, TSS clusters identified by TSS-seq in this study with start position (start), end position (stop), and the position which is supported by the highest number of TSS-seq read within that cluster (max), the strand of the gene, and the annotation of TSS clusters as "Annotated" or "Alternative" as in Methods.
(XLSX)

**S5 Table. List of genes for which the condition-dependent regulation of altTSS usage results in the production of protein isoforms with different subcellular localization as predicted by Deeploc2.0.**
(XLSX)

**S6 Table. N-terminomic analysis.** (A) Peptides identified by N-terminomic. (B) Peptides specific of mRNAs resulting from altTSS usage identified by N-terminomic.
(XLSX)

**S7 Table. List of genes regulated by altTSS in response to change in temperature or/and growth phase.** GeneID of genes identified as regulated by alternative TSS usage in this study, followed by TSS clusters, their positions, and their normalized expression in each replicates as described in Methods, mean normalized expression in each condition, difference in mean normalized expression, and false discovery rate of the differential comparison (FDR).
(XLSX)

**S8 Table. List of genes regulated by the exponential to stationary phase transition in the WT and *tur1Δ* in *C. neoformans*.**
(XLSX)

**S9 Table. List of genes with at least 2 TSSs and at least 20 TSS reads in all 4 samples (WT E30, WTS30, *tur1Δ* E30, and *tur1Δ* S30).** GeneID of genes identified as regulated by alternative TSS usage in this study, followed by TSS clusters, their positions, and their normalized expression in each replicates as described in Methods, mean normalized expression in each condition, difference in mean normalized expression, and false discovery rate of the differential comparison (FDR).
(XLSX)

**S10 Table. S10 Table positions and weight to the peaks identified by ChIP-seq analysis.**
(XLSX)

**S11 Table. Position of the methylated adenine residues identified by DamID-Seq and specific of either stationary or exponential phase.**
(XLSX)

**S1 Fig. TSS cluster sizes obtained after directly merging the 12 gff files originally generated in Wallace and colleagues [14].**
(DOCX)

**S2 Fig. TSS clusterization using *C. neoformans* TSS-seq data obtained in 3 other growth conditions also revealed 2 types of TSS (broad and sharp).** (A) Stationary phase 30˚C (B) exponential phase 37˚C, and (C) stationary phase 37˚C. Left. Distribution of the TSS clusters in terms of cluster width (size) and shape index (SI) is represented as a 2D density plot in which size and SI distribution of TSS clusters display a bimodal pattern. Corresponding histogram of size and SI are projected on x-axis and y-axis, respectively. Right. Histogram (bold blue line) and density plot (bold black line) of the first principal component (PC1) of size and SI. Two subpopulations of PC1 are detected by statistical test using "MixtureInf" R package and represented as the theoretical density plot (dotted black line). The data underlying this figure can be found in S1 Data.
(DOCX)

**S3 Fig. Similar motif enrichment was observed associated with broad and sharp clusters generated from each set of TSS-seq data.** Enrichments and positions relative to the major

position within the TSS cluster of 6 detected motifs within the TL sequence.
(DOCX)

**S4 Fig. Mae102 localization under exponential phase.** The number of cells with Mea102 nuclear localization is indicated.
(DOCX)

**S5 Fig. altTSS usage regulation of *SOD1* and *SOD2* genes in exponential and stationary phases.** IGV visualization of RNA-seq (upper tracks) and TSS-seq (bottom tracks) at the CNAG_01019 (*SOD1*) and CNAG_04388 (*SOD2*) loci when *C. neoformans* cells were cultivated at 30˚C under exponential or stationary phase, respectively.
(DOCX)

**S6 Fig. Apparent "non-conserved" altTSS can be conserved between *C. neoformans* and *C. deneoformans*.** IGV visualization of RNA-seq and TSS-seq at the *VPS70* and *PKP1* loci of *C. neoformans* and *C. deneoformans* obtained when cells were cultivated at 30˚C under stationary phase. At the *C. deneoformans VPS70* gene, the same altTSS observed in *C. neoformans* is visible albeit very poorly used. At the *C. deneoformans PKP1* gene, an altTSS is also visible but not at the same position as the one observed in *C. neoformans*. Interestingly, in both cases, the regulation of altTSS is reversed between species.
(DOCX)

**S7 Fig. Sensitivity of the *tur1Δ* mutant to oxidative agents.** Ten-fold serial dilutions (starting with $10^7$ cells/ml) of the wild type (WT), *tur1* mutant (*tur1Δ*), and complemented strain (*tur1Δ TUR1*) cells were spotted on YPD medium containing Cumen hydroperoxide (CHP), $NaNO_2$, sodium dithiocarbamate, rotenone, dimethyl malonate, or oligomycin. Plates were incubated 3 days at 30˚C before being photographed.
(DOCX)

**S8 Fig. Optimization of TSS data clusterization.** (A) Left panel: Cumulative percentage of clusters that reach size when testing a range of d values from 1 to 50. Right panel: Correlation between d and percentage of clusters that reach a maximum size. The red rectangle illustrates the areas where d runs from 12 to 18 in both panels. (B) Left panel: The data in (A) is modelized as the red smooth curve. Right panel: Correlation between d and modelized percentage of clusters that reach maximum size, which reaches the maximum value at d = 17 (red line). The data underlying this figure can be found in S1 Data.
(DOCX)

**S1 Raw Images. Uncropped images corresponding to Figs 6B and 8C.**
(PDF)

**S1 Data. Raw data used to draw the Figs 1, 2, 3, 7, 8, 9, 11B, 12, S2, S3 and S8.**
(ZIP)

## Author Contributions

**Conceptualization:** Guilhem Janbon.

**Data curation:** Ida Chiara Guerrera.

**Formal analysis:** Thi Tuong Vi Dang.

**Funding acquisition:** Guilhem Janbon.

**Investigation:** Thi Tuong Vi Dang, Corinne Maufrais, Frédérique Moyrand, Isabelle Mouyna, Jean-Yves Coppée, Chinaemerem U. Onyishi, Joanna Lipecka.

**Project administration:** Guilhem Janbon.

**Supervision:** Jessie Colin, Ida Chiara Guerrera, Robin C. May, Guilhem Janbon.

**Validation:** Isabelle Mouyna.

**Writing – original draft:** Thi Tuong Vi Dang.

**Writing – review & editing:** Jessie Colin, Robin C. May, Guilhem Janbon.

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
