## [Editor Report · Decision Letter 0]

31 Jul 2023

Dear Dr. Janbon, 

Thank you for submitting your manuscript entitled "Tur1 regulates alternative TSS usage in Cryptococcus" for consideration as a Research Article by PLOS Biology.

Your manuscript has now been evaluated by the PLOS Biology editorial staff, as well as by an academic editor with relevant expertise, and I am writing to let you know that we would like to send your submission out for external peer review.

Once your full submission is complete, your paper will undergo a series of checks in preparation for peer review. After your manuscript has passed the checks it will be sent out for review. To provide the metadata for your submission, please Login to Editorial Manager (https://www.editorialmanager.com/pbiology) within two working days, i.e. by Aug 02 2023 11:59PM.

Kind regards,

Paula

---

Senior Editor

PLOS Biology

---

## [Decision Letter · Decision Letter 1]

6 Oct 2023

Dear Dr Janbon,

Thank you for your patience while your manuscript "Tur1 regulates alternative TSS usage in Cryptococcus" was peer-reviewed at PLOS Biology. It has now been evaluated by the PLOS Biology editors, an Academic Editor with relevant expertise, and by several independent reviewers. 

In light of the reviews, which you will find at the end of this email, we would like to invite you to revise the work to thoroughly address the reviewers' reports.

As you will see below, the reviewers agree that the work is interesting, but further evidence to support your conclusions is needed. In particular, we think that it is important that you explain and underscore more specifically the potential impact of TSS usage changes on Cryptococcus biology, especially infection biology an/or mating. Reviewers #2 and #3 think that indirect effects of Tur1 are an issue. We consider that you should provide more compelling insight into what Tur1 does. We suggest to look at binding at or around TSS's by ChIP-Seq, and the association with RNA Polymerase. Please address the rest of the reviewers' reports.

Given the extent of revision needed, we cannot make a decision about publication until we have seen the revised manuscript and your response to the reviewers' comments. Your revised manuscript is likely to be sent for further evaluation by all or a subset of the reviewers.

**IMPORTANT - SUBMITTING YOUR REVISION**

*Re-submission Checklist*

*Published Peer Review*

*PLOS Data Policy*

*Blot and Gel Data Policy*

Sincerely,

Paula

---

Senior Editor

PLOS Biology

REVIEWS:

Reviewer #1: Cryptococcus environmental adaptation.

Reviewer #2: Cell fate and gene regulation in yeast.

Reviewer #3: Novel translational control elements.

Reviewer #1: The authors performed genome-wide analysis for transcriptional start sites in two related Cryptocoocus species. This study builds upon a recent work by the same group in which they mapped RNA transcripts versus the associated proteins, demonstrating a large number of upstream ORFs associated with inhibition of translation. They also demonstrated specific examples in which alternative transcript lengths resulted in altered protein localization, often by removing mitochondrial localization sequences.

In the current paper, the authors systematically and comprehensively define transcription start sites (TSS) for these Cryptococcus species, identifying thousands of alternative TSS. They further categorized TSS as occurring in the region prior to annotated ATGs, or further downstream, thus altering protein sequence. They were also able to identify condition-dependent alterations in TSS usage, providing strong evidence that this is a major means of protein regulation in this pathogenic fungus. Similar to their prior study, but on a much larger scale, they showed that some of these protein sequence changes resulted in altered localization.

Some of the general themes that they explore and define are consistent with paradigms from other species (e.g. S pombe). These include the observation that the present of TATA sequences in promoters is correlated with more dynamic, condition-specific, transcriptional regulation. However, the rigor with which these observations were defined is important to emphasize. Moreover, the relevance of these studies to host-pathogen interaction offers a very unique position for this paper relevant to other literature on this topic.

The logic of the experimental strategy is very clear, and the analysis is very rigorous. Data have been made available in accessible databases. 

Minor issues.

1) The presentation is mostly very clear. In one instance (Fig 6), however, the resolution of the protein localization was not completely evident by the provided images.

2) The authors do not adequately explain the motifs identified by the MEME algorithm in the Figure 1C. How is this informative for the larger story? This seems to be an opportunity that is missed.

3) Is the phrase "TATA-box containing genes" accurate since the sequence typically resides outside the ORF?

4) The authors could better describe how they developed their parameters for a TSS cluster to be "in close proximity" to the annotated ATG versus "far downstream". Is associated Fig 3 necessary?

5) There are some instance of excessive descriptives, including e.g, "spectacular effect" (line 467); "… suggest a fantastic regulatory …" (line 588).

6) Line 535. "… no growth defect of the tur1 strain …"

7) Line 596. "… examples exist in model yeasts …"

Reviewer #2: The authors characterise TSS usage in two species of pathogenic Cryptococcus, and define different TSS cluster characteristics (sharp and broad). They then look at differential TSS usage between different environmental conditions, characterising the effects based on the location of the aTSS with respect to the protein's annotated translation start codon. They describe an example gene where the subcellular localisation is supposedly altered by aTSS usage (resulting in the inclusion or exclusion of a localisation signal). They next look at the molecular mechanisms underpinning TSS usage and identify a TF (Tur1) which regulates downstream TSS usage in the stationary phase. They characterise the phenotype of Tur1 mutant strains. 

Overall the manuscript is quite descriptive, often highlighting examples without further in detail characterization. Also the manuscript lacks some level of coherency. Terms are often poorly defined or not defined at all (e.g. alternative TSS). Moreover, often descriptions of how cut-offs are made are not included in main text making it difficult to appreciate the findings. 

Specific main comments:

- The title of the manuscript does not really match the manuscript as only the last couple of figures are on Tur1, and Tur1 has an effect on subset of TSSs but not genome wide. 

- It was not clear to me what the definition of alternative TSS entails. Based on reading the text it seems all genes that have 2 TSSs or more? It would be good to give a clear definition what it is alternative to. The literature often discriminates between main vs alternative TSS usage or dominant TSS vs weak TSS. In figure 4BC which of the TSSs is the alternative one?

- Figure 6 the authors attempt to demonstrate a difference in localization. However this was difficult to assess. The images are of poor quality, and no merge was shown. Also no quantifications were shown, and no mitotracker for exp was shown. 

- Tur1 effect on TSS usage looks interesting. However, it was not clear to me whether this was direct or indirect effect on TSS usage. Given that Tur1 also has a phenotype indirect effects are plausible. To discriminate between direct and indirect at least ChIP-seq of Tur1 is needed. 

minor comments.

- manuscript would benefit from thorough read through spelling and acronym errors.

- proper definitions of terms used. E.g. Figure 3 what are the cut-off used.

---

## [Editor Report · Decision Letter 2]

6 Jun 2024

Dear Dr Janbon,

Thank you for your patience while we considered your revised manuscript "Alternative Transcription Start Site usage in Cryptococcus" for consideration as a Research Article at PLOS Biology. Your revised study has now been evaluated by the PLOS Biology editors and the Academic Editor. 

In light of the reviews, which you will find at the end of this email, we are pleased to offer you the opportunity to address the remaining points (see below) from the Academic Editor in a revision that we anticipate should not take you very long. We will then assess your revised manuscript and your response with our Academic Editor aiming to avoid further rounds of peer-review.

IMPORTANT - please attend to the following requests from the Academic Editor:

a) The Academic Editor strongly recommends that you present some kind of model that connects Tur1 to start site selection, at least at the PKP1 gene. You can call the model tentative; you can provide two (but not more) models; but they must be specific, not just hand-waving. Acceptance will depend upon the clarity and logic of your new Tur1-Tss1 explanation and compliance with requests below.

b) Several supplementary files are simply computer program outputs that are incomprehensible to an average reader. Each supplementary table needs a legend that explains what the columns are; otherwise the files are useless. Your most enthusiastic readers will want to use your data to find genes that they are interested in, but right now they will be unable to do so. In short, the files are not for documentation; they are for utilization.

c) Line 252 - the term "spectacular" should be toned down as per Rev 1's previous recommendations.

d) Figure 5B The terms "Stationary" is misspelled

**IMPORTANT - SUBMITTING YOUR REVISION**

*Resubmission Checklist*

*Published Peer Review*

*PLOS Data Policy*

*Blot and Gel Data Policy*

Sincerely,

Melissa

Melissa Vazquez Hernandez, Ph.D.

Associate Editor

PLOS Biology

---

## [Editor Report · Decision Letter 3]

17 Jun 2024

Dear Dr Janbon,

Thank you for your patience while we considered your revised manuscript "Alternative Transcription Start Site usage in Cryptococcus" for publication as a Research Article at PLOS Biology. This revised version of your manuscript has been evaluated by the PLOS Biology editors, the Academic Editor.

Based on our Academic Editor's assessment of your revision, we are likely to accept this manuscript for publication, provided you satisfactorily address the remaining points raised below. Please also make sure to address the following data and other policy-related requests.

a) We would like to suggest the following modification to the title:

"Alternative TSS use is widespread in Cryptococcus fungi in response to environmental cues and regulated genome-wide by the transcription factor Tur1"

b) You stated that there was no financial support. Please make sure this is the case.

c) As we requested before, the supplementary tables should contain legends that describe what do the column headers mean, and what is being shown in each table. For example, in table S11, there is no legend and the column headers read “gneId name pos DAM_EXPO_C DAM_EXPO_B DAM_EXPO_A DAM_STAT_C DAM_STAT_B DAM_STAT_A”, which without a proper explanation will be hard for the readers to understand their meaning. Also be aware that some titles have the word “pics” which might mean “peak”.

Please supply the numerical values either in the a supplementary file or as a permanent DOI’d deposition for the following figures:

Figure 1ACDE, 2ABC, 3A, 8AB, 9A, 11A, 12B, S2ABC, S8

e) Please cite the location of the data clearly in all relevant main and supplementary Figure legends, e.g. “The data underlying this Figure can be found in S1 Data” or “The data underlying this Figure can be found in https://doi.org/10.5281/zenodo.XXXXX”

f) We require the original, uncropped and minimally adjusted images supporting all blot and gel results reported in the following Figures:

Figure 6B, 8C

We will require these files before a manuscript can be accepted so please prepare and upload them now. Please carefully read our guidelines for how to prepare and upload this data: https://journals.plos.org/plosbiology/s/figures#loc-blot-and-gel-reporting-requirements

g) Please ensure that your Data Statement in the submission system accurately describes where your data can be found and is in final format, as it will be published as written there.

h) Per journal policy, if you have generated any custom code during the curse of this investigation, please make it available without restrictions upon publication. Please ensure that the code is sufficiently well documented and reusable, and that your Data Statement in the Editorial Manager submission system accurately describes where your code can be found.

We expect to receive your revised manuscript within two weeks. 

*Published Peer Review History*

*Press*

Sincerely,

Melissa

Melissa Vazquez Hernandez, Ph.D.

Associate Editor

PLOS Biology

---

## [Editor Report · Decision Letter 4]

28 Jun 2024

Dear Dr Janbon,

Thank you for the submission of your revised Research Article "Alternative TSS use is widespread in Cryptococcus fungi in response to environmental cues and regulated genome-wide by the transcription factor Tur1" for publication in PLOS Biology. On behalf of my colleagues and the Academic Editor, Aaron Mitchell, I am pleased to say that we can in principle accept your manuscript for publication, provided you address any remaining formatting and reporting issues. These will be detailed in an email you should receive within 2-3 business days from our colleagues in the journal operations team; no action is required from you until then. Please note that we will not be able to formally accept your manuscript and schedule it for publication until you have completed any requested changes.

PRESS

Sincerely, 

Melissa 

Melissa Vazquez Hernandez, Ph.D., Ph.D.

Associate Editor

PLOS Biology
